# Development of multifunctional membranes via plasma-assisted nonsolvent induced phase separation

Yueh-Han Huang [1], Meng-Jiy Wang [2] & Tai-Shung Chung [1,2,3] ✉

Demands on superhydrophobic, self-cleaning and piezoelectric membranes have gained significantly due to their potential to overcome global shortages in clean water and energy. In this study, we have discovered a novel plasma-assisted nonsolvent induced phase separation (PANIPS) method to prepare superhydrophobic, self-cleaning and piezoelectric poly(vinylidene difluoride) (PVDF) membranes without additional chemical modifications or post-treatments. The PANIPS membranes exhibit water contact angles ranging from 151.2° to 166.4° and sliding angles between 6.7° and 29.7°. They also show a high piezoelectric coefficient (d33) of 10.5 pC N$^{-1}$ and can generate a high output voltage of 10 V$_{pp}$. The PANIPS membranes can effectively recover pure water from various waste solutions containing Rose Bengal dye, humic acid, or sodium dodecyl sulfate via direct contact membrane distillation (DCMD). This study may provide valuable insights to fabricate PANIPS membranes and open up new avenues to molecularly design advanced superhydrophobic, self-cleaning, and piezoelectric membranes in the fields of clean water production, motion sensor, and piezoelectric nanogenerator.

The impacts of global warming and industrialization have resulted in an increased demand for clean water and energy[1]. Membrane technologies are recognized as a sustainable and environmentally friendly solution to address the issues of water and energy scarcity, offering several advantages over alternative methods, including ease of use, flexibility, and adaptability[2–4]. It is reported that the surface characteristics of membranes, particularly their wettability, play a crucial role in their performance[5,6]. Recently, there has been a growing interest in superhydrophobic and self-cleaning membranes with water contact angles (WCAs) > 150° and sliding angles (SAs) < 10°[7–9] due to their superior performance in various applications. For example, non-wetting superhydrophobic membranes can prevent the attachment of liquid absorbents, ensuring a stable $CO_2$ absorption flux in membrane contactors[10,11]. In membrane distillation (MD), the superhydrophobic membranes can enhance the vapor flux and effectively mitigate the issues such as scaling, fouling, and wetting[12–15].

The wettability of a membrane is influenced by several factors, but when it comes to fabricating superhydrophobic membranes, two key parameters are emphasized: low surface energy and high roughness[16,17]. Following these principles, researchers have developed various superhydrophobic membranes via extrinsic modifications or intrinsic property alteration, as listed in Supplementary Table 1. Extrinsic modifications involve applying coatings or surface treatments on conventional membranes to enhance their superhydrophobic properties. Coating materials like carbon nanotubes, ZnO nanorods, and fluorinated $SiO_2$ nanoparticles have been utilized to mimic the nature inspired hierarchical structures found on lotus leaves[18–21]. Additionally, superhydrophobic surfaces can be achieved by different plasma techniques, such as plasma polymerization, plasma etching, and plasma treatment[22–25]. For example, membranes treated by $CF_4$ plasma show an enhanced wetting resistance due to the plasma assisted fluorination and the plasma etching effect[26]. Extrinsic

[1]Graduate Institute of Applied Science and Technology, National Taiwan University of Science and Technology, Taipei 106335, Taiwan. [2]Department of Chemical Engineering, National Taiwan University of Science and Technology, Taipei 106335, Taiwan. [3]Department of Materials Science and Engineering, National Taiwan University of Science and Technology, Taipei 106335, Taiwan. ✉e-mail: chencts@mail.ntust.edu.tw

modifications offer the advantages of being applicable to existing membrane materials, allowing for easy retrofitting of membranes to achieve superhydrophobicity. However, the deposition and modification procedures could be both time- and chemical-consuming. Additionally, there is a risk of leaching the deposited components from the membranes, which not only compromises their superhydrophobicity but also raises concerns about potential environmental toxicity[27].

On the other hand, the hydrophobicity of membranes can be intrinsically altered by changing their surface structure to impart higher hydrophobicity. The membrane structure can be adjusted by controlling the phase separation behavior during membrane formation, using common methods such as nonsolvent-induced phase separation (NIPS), vapor-induced phase separation (VIPS), and electrospinning. In the NIPS method, a polymer dope solution is cast onto a glass plate and then transferred to a nonsolvent coagulant bath. By using soft coagulants (e.g., alcohols), the delayed liquid-liquid demixing occurs and promotes the membrane roughness and porosity, thereby significantly increasing the membrane's hydrophobicity[28,29]. Although using soft coagulants in NIPS can create a more hydrophobic membrane, the usage of massive organic solvents is a major concern. On the other hand, in the VIPS method, the as-cast membrane is exposed to humid air for a certain period before transferring to the coagulant bath. The absorbed vapor gradually induces phase separation through solid-liquid demixing, resulting in the formation of larger polymer crystals and thus increasing the hydrophobicity of the membrane[30]. The VIPS method takes a longer processing time compared to the NIPS method. In addition, no matter using a soft coagulant in NIPS or prolonging the exposure time in VIPS, the membranes would have a rougher and more porous surface, but their WCAs are difficult to reach the level of superhydrophobicity, as tabulated in Supplementary Table 1[31–34]. Therefore, additional coating with low-surface-energy materials are still required. The electrospun nanofiber membranes inherently have a higher roughness than flat sheet membranes due to their fibril structure, thereby possessing a higher WCA. However, it is also difficult to achieve superhydrophobicity via electrospinning. Lie et al. demonstrated the electrospun PVDF membrane had a WCA of 146°, which was further increased to 171.5° by adding 8% ZnO in the electrospun solution[35].

Recently, Lu et al. proposed a more effective rheological spray-assisted nonsolvent induced phase separation (SANIPS) method to fabricate the superhydrophobic and self-cleaning membranes. In the SANIPS method, the as-cast membrane is first sprayed with designated materials such as air, water, or ethanol for 30 ~ 60 s, and then immersed in a coagulant bath to complete the phase inversion[36]. The physical impact of the compressed air flow or liquid droplets induces local distortion of the membrane surface, thus enhancing the roughness. At the same time, the spraying also expedites the moisture condensation that accelerates the solid-liquid demixing and forms a porous skin layer. After transferring it into a coagulant bath, the skin layer retards the liquid-liquid demixing and suppresses the formation of macrovoids[37]. Using the SANIPS method, PVDF membranes exhibit superhydrophobicity and self-cleaning without any post-treatment.

Inspired by the SANIPS method, herein we present a facile and green method to fabricate superhydrophobic and self-cleaning membranes via a PANIPS method. Unlike traditional plasma techniques that usually applied vacuum plasma on existing membrane materials, an atmospheric pressure Ar microplasma jet is applied on the as-cast membranes to manipulate their microstructure and morphology (see Fig. 1). PVDF, a commonly used and well-studied semicrystalline ferroelectric polymer, is chosen as the model polymer in this work. The evolution of physicochemical properties of the PANIPS PVDF membranes, such as morphology, roughness, porosity, water contact angle (WCA), sliding angle (SA), and crystalline phase, is systematically investigated and correlated with the key plasma parameters. Moreover, the underlying mechanisms are studied and discussed to understand the interaction between plasma and the nascent membranes. Finally, the performances of the PANIPS membranes are examined by direct contact membrane distillation (DCMD) and reciprocating piezoelectric output voltage tests to validate their potential for clean water production, motion sensor, and piezoelectric nanogenerator applications.

## Results

### Morphology and topography of PANIPS membranes

The membrane morphologies and topographies with and without plasma treatment are displayed in Fig. 2. The NIPS membrane possesses a typical asymmetric structure comprising a dense skin layer and macrovoids beneath the top surface (Fig. 2 (a1), (b1) and (c1)) due to the fast liquid-liquid demixing and the nonsolvent intrusion during the phase inversion process[38]. On the other hand, microplasma treatments impose significant changes in the membrane morphology and structure. As revealed by the SEM images, many visible pores appear after scanning for 1 cycle, and the size of macrovoids also shrinks significantly (Fig. 2 (a2), (b2) and (c2)). With 3 scan cycles, the dense skin layer develops to particulate PVDF crystals, and the membrane becomes macrovoid-free (Fig. 2 (a3), (b3) and (c3)). A further increase

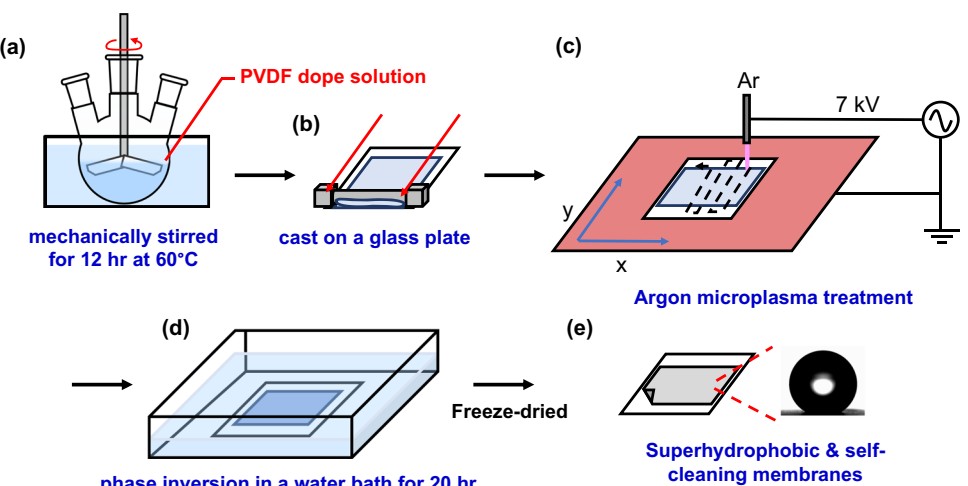

**Fig. 1 | The schematic illustration of the experimental procedures of PANIPS membrane preparation comprising. a** dope preparation (12% PVDF, 10% EG, and 78% NMP), **b** membrane casting on a glass plate using a casting knife with a gap height of 300 μm, **c** Ar microplasma treatment, **d** phase inversion in tap water, and **e** resultant superhydrophobic and self-cleaning PVDF membranes.

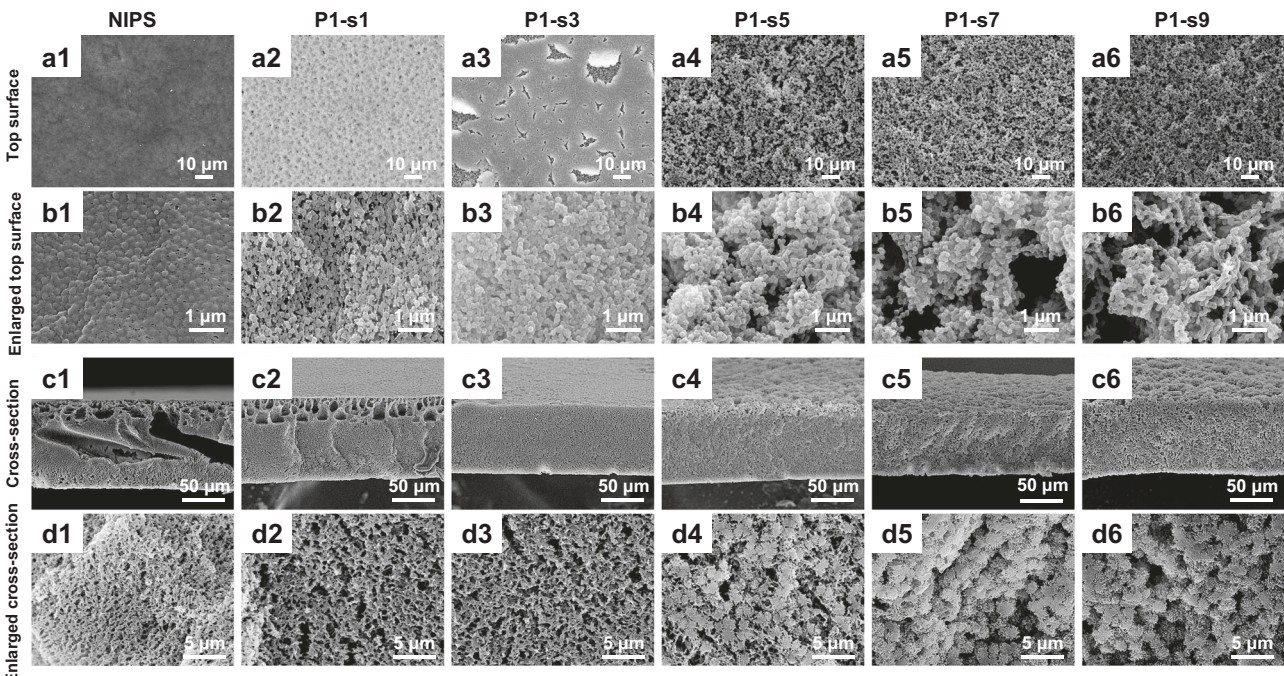

**Fig. 2 | The surface morphologies and structures of PANIPS membranes. a** Top surface and **b** enlarged top surface, **c** cross-section, and **d** enlarged cross-section SEM images of NIPS and PANIPS PVDF membranes: (1) NIPS, (2) P1-s1, (3) P1-s3, (4) P1-s5, (5) P1-s7, and (6) P1-s9 membranes.

in plasma scan cycle from 5 to 9 makes the surface fully porous and creates multilevel particulate PVDF polymer crystals that significantly increase the surface roughness (Fig. 2 (a4-a6)). Moreover, plasma scanning for 5 cycles (P1-s5) seems to be the critical point for the transition of cross-sectional structure from an interconnected bicontinuous structure to a separated spherulitic structure.

When the as-cast dope solution was treated by microplasma, it was observed that the membrane turned to opaque at more scan cycles, implying the plasma treatment was able to induce partial phase separation at the membrane surface. The produced skin layer after plasma treatment could be clearly observed by a microscope. As shown in Supplementary Fig. 1, the as-cast membrane is fully transparent. After immersing in water for 10 s (NIPS), the membrane shows a rough mountain-valley structure because of the rigorous phase separation. After plasma treatment (P1-s9), the skin layer appears, indicated by the abundant black dots that are homogeneously distributed on the surface. In comparison, a surface skin layer is also observed on the VIPS membrane prepared with the same exposure time in humid air, but its roughness is apparently much less than that treated by plasma. It is speculated that the longer plasma treatment time, the thicker skin layer is formed. Thus, after immersing in the coagulant bath, the porous skin layer may retard the solvent-nonsolvent exchange and the evolution of membrane structure is the outcome of the delayed demixing[36].

### Superhydrophobicity and self-cleaning properties of PANIPS membranes

To verify whether the plasma treatment strengthens the hydrophobicity of the PVDF membranes, the WCAs and SAs of the PVDF membranes were measured. As shown in Fig. 3a, the NIPS membrane possesses a moderate hydrophobicity with a WCA of 95°. In contrast, the hydrophobicity of the PVDF membranes is significantly improved by plasma treatment. Their WCA values increase to 129°, 151°, 159°, 160°, and 164° after plasma treatment for 1, 3, 5, 7, and 9 cycles, respectively. It is worth noting that although the PVDF membrane becomes superhydrophobic (WCA > 150°) with 3 scan cycles, water droplets still adhere to the membrane even at a tilting angle of 90°. It

takes 5 scan cycles for water droplets to start rolling off, and more than 7 scan cycles to achieve a SA lower than 10°.

It is well-known that minimizing the contact area is a key to allowing a liquid droplet to retain its Cassie-Baxter state on a solid surface with the microscopic air pockets trapped below the liquid phase[39]. In addition, recent studies have emphasized the positive effects of the re-entrant structure in promoting the self-cleaning properties of composite interfaces[17]. As a result, the changes in roughness and the surface porosity were measured to correlate the enhanced superhydrophobicity and the membrane morphology. As presented in Fig. 3b and Supplementary Fig. 2, it is noticed that both the roughness and surface porosity drastically increase and reach a plateau after 5 scan cycles. Then, the droplets start to slide down from the membrane surface. These results imply that the low SA and the superhydrophobicity are ascribed to the enhanced roughness and surface porosity because of the plasma treatment.

The movement of water droplets on both the pristine NIPS membrane and plasma treated membrane (P1-s9) is displayed in Fig. 3c. When a water droplet is dropped onto the 10°-tilted NIPS membrane, it spreads out and adheres firmly to the surface. By contrast, the water droplet bounces and rolls off immediately from the P1-s9 membrane, showing its outstanding self-cleaning property. The plasma treated membranes also exhibit resistance to a variety of aqueous solutions, including 0.2 mM sodium dodecyl sulfate (SDS), 500 ppm Rose Bengal, or 10% ethanol solutions (Fig. 3d), highlighting their potential for the use in treating versatile wastewaters. To demonstrate the improved self-cleaning ability of P1-s9, both NIPS and P1-s9 membranes were immersed in a 500 ppm Rose Bengal dye solution. Upon removal from the solution (Fig. 3e), the P1-s9 membrane showed no residual solution, in great contrast to the pristine NIPS membrane which became visibly pink. The improved self-cleaning characteristic of PANIPS membranes can be primarily attributed to their increased surface roughness and porosity, which function similarly to the lotus effect in maintaining the cleanliness of the membrane surface[39].

The effects of plasma treatment parameters such as working distance and plasma scan speed on membrane wettability are further

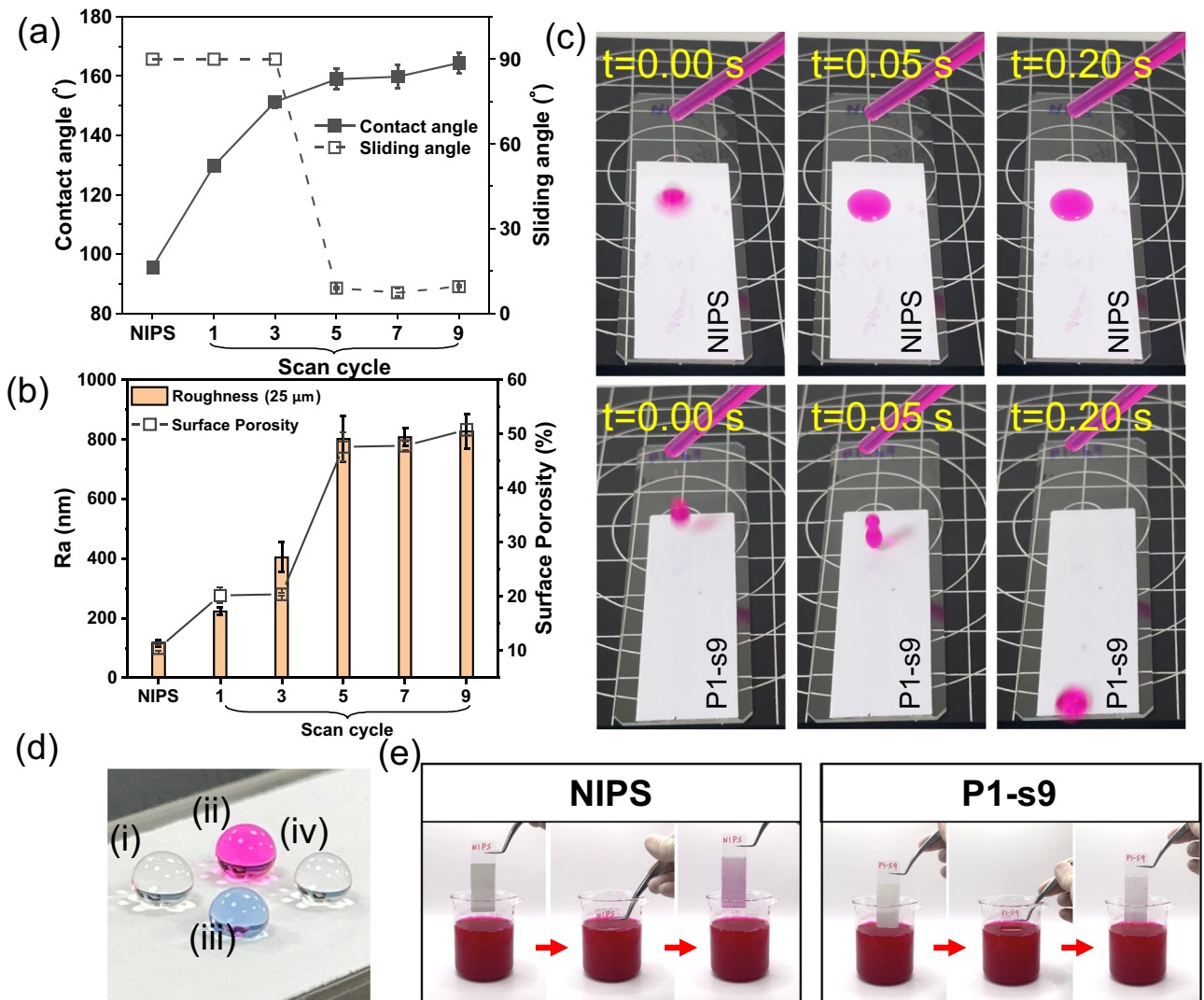

**Fig. 3 | The wettability analyses of the PANIPS membranes. a** Contact angles and sliding angles. Data were presented as the mean of 4 replicates ± standard deviation. **b** Surface roughness and porosities of PVDF membranes. Data were presented as the mean of 3 replicates ± standard deviation. **c** Images showing Rose Bengal dyed water droplets being dropped on NIPS and P1-S9 membranes at intervals of 0-0.2 s. Membranes were tilted at a 10° angle. **d** Images of different droplets on the P1-S9 membrane: (i) DI-water, (ii) 500 ppm Rose Bengal dyed water, (iii) 0.2 mM SDS, and (iv) 1:9 ethanol/water mixture. **e** Self-cleaning tests of NIPS and P1-S9 membranes by immersing them in a 500 ppm Rose Bengal dye solution.

investigated and the details are discussed in Supplementary Note 1 and Supplementary Fig. 3.

## Membrane characterizations

PVDF is a semi-crystalline polymer known for its high dielectric constant and electroactive response. Its molecular structures and crystalline phases such as $\alpha$-, $\beta$-, and $\gamma$-phases have been well investigated[40]. As displayed in Fig. 4a, the $\alpha$-phase, which is widely observed in NIPS membranes, is not electroactive due to antiparallel packing of the dipoles within the unit cell. On the other hand, $\beta$- and $\gamma$-phases are the most electroactive phases in PVDF, attributed to their strong dipole moment perpendicular to the polymer chains[41]. The crystalline phases of both NIPS and plasma-treated membranes were characterized by ATR-FTIR and XRD. As shown in Fig. 4b, the characteristic peaks of $\alpha$- and $\beta$-phase PVDF at 766 cm$^{-1}$ and 1279 cm$^{-1}$, respectively, are observed in NIPS, P1-s1, and P1-s3 membranes[40]. When the scan cycle is more than 5, it is intriguing to observe that the characteristic peaks of $\alpha$-phase vanish. Instead, a new peak at 1235 cm$^{-1}$ exclusively corresponding to the electroactive $\gamma$-phase appears, indicating the $\alpha \to \gamma$ transition is induced by the plasma treatment[40]. Consistent with IR

analyses, XRD shows that NIPS, P1-s1, and P1-s3 have two strong diffraction peaks at 18.4° and 20.0°, corresponding to 020 and 110 reflections of the monoclinic $\alpha$-phase crystal, respectively (Fig. 4c)[42]. When the plasma scan cycle is greater than 5, both peaks shift to 18.5 and 20.3°, respectively, which belong to the planes 020 and 110/101 of the $\gamma$-phase crystal, separately[42]. In addition, a shoulder peak at 20.6° ascribed to the reflection of $\beta$-phase crystal is observed in all membranes. Both the IR and XRD results confirm that plasma treatment for > 5 cycles converts PVDF from a mixture of $\alpha$- and $\beta$-phases to the fully electroactive $\beta$- and $\gamma$-phases, which would largely promote the electroactivity of PVDF membranes. It is observed that such phase transition not only varies with the scan cycle but also with the working distance. As summarized in Supplementary Fig. 4, increasing the working distance to 2.5 cm shows a similar $\alpha/\beta \to \beta/\gamma$ transition when the scan cycle is > 5. However, the phase transition is less pronounced when the distance is further increased to 5 cm, and it shows no obvious change in the crystalline phase when the distance is 10 cm. It is important to highlight that the PANIPS process exclusively alters the crystalline polymorphisms and morphology of the PANIPS membranes without introducing new chemical bonding. As revealed by the XPS

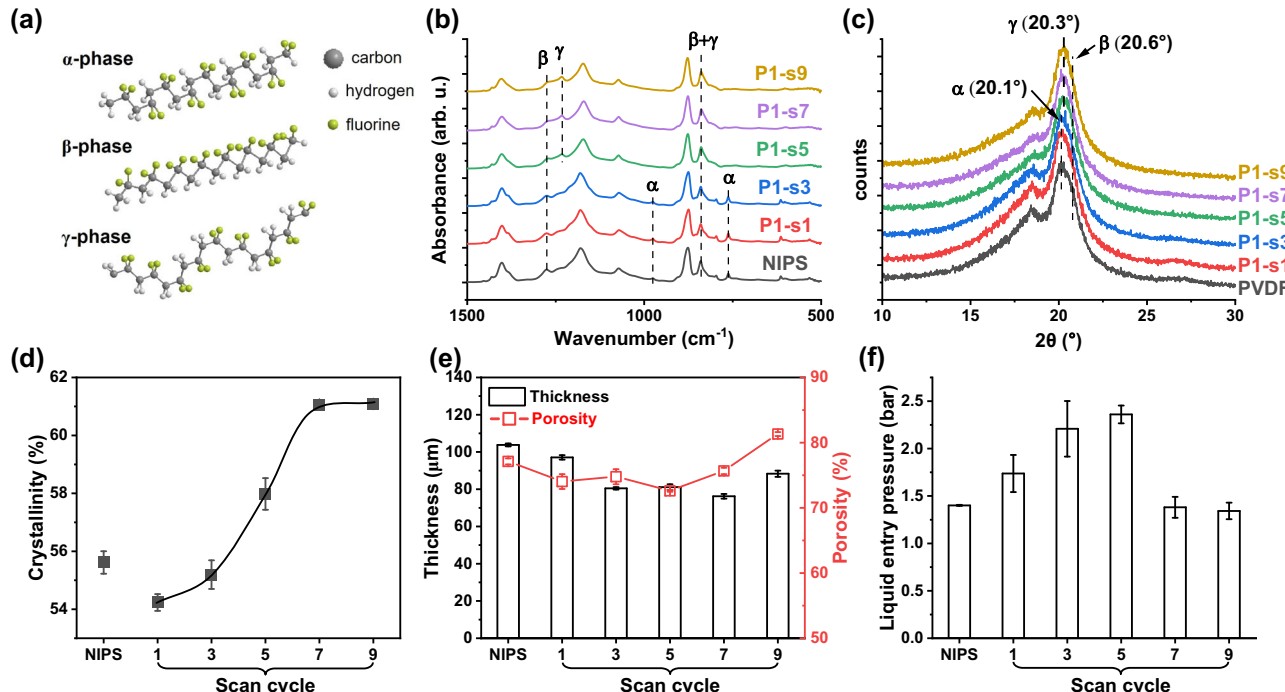

**Fig. 4 | Characterizations of the PANIPS membranes. a** The molecular structures of a, b, and g-phase PVDF, **b** FTIR spectra, **c** XRD, **d** crystallinity, **e** thickness and porosity, and **f** liquid entry pressures. Plasma treatment was all carried out at the working distance 1 cm, 7 KV, 20 kHz. The humidity and temperature were RH 70% and 25 °C, respectively. Data in (**d**–**f**) were presented as the mean of 3 replicates ± standard deviation.

survey spectra, the NIPS and PANIPS membranes only display characteristic peaks of C1*s* and F1*s* (Supplementary Fig. 5a). A closer examination of the C1*s* narrow scan spectra further confirms that all membranes share common peaks of C-H (286.6 eV) and C-F (291.2 eV) with equivalent peak intensities (Supplementary Fig. 5b).

The crystallinity of the PANIPS membranes was investigated by DSC. As shown in Fig. 4d, the NIPS membrane has a crystallinity of 55.6%. It is observed that short scan cycles of 1 and 3 result in slightly lower crystallinities of 54.2% and 55.2%, respectively. As the scan cycle increases from 5 to 9, the membrane crystallinity continuously increases to 61.1%. As mentioned in Section 3.1, the plasma treatment induces a partial phase separation and forms a porous and hierarchical skin layer that effectively prevents the formation of macrovoids due to the delayed demixing. However, the porous skin layer at short scan cycles remains thin and unstable, therefore ineffective in inducing the delayed demixing. Instead, the more pores at the surface provide more channels for water intrusion, making the membrane quickly precipitated by liquid-liquid demixing. Therefore, the crystallinities of P1-s1 and P1-s3 membranes are slightly lower than that of the NIPS membrane due to a shorter crystallization time. On the other hand, the thicker skin layer formed at longer scan cycles (P1-s5 ~ P1-s9) can retard the solvent-nonsolvent exchange. In this case, the nascent PVDF membrane has more time for nucleation and crystallization because of the delayed demixing, hence achieving a higher crystallinity. A similar phenomenon is also observed in the membranes prepared by the VIPS method. In VIPS membranes, with a sufficient exposure time in humid air, PVDF has more time to crystallize due to the slow mass transfer between the solvent (NMP) and nonsolvent (i.e., water intake from humid air). Therefore, the crystal size and the total crystallinity usually increase with a longer exposure time[43].

Figure 4e shows the thickness and porosity of the PANIPS membranes. The NIPS membrane has a thickness of 104 μm. In contrast, the thickness of PANIPS membranes decreases from 97 μm (P1-s1) to 76 μm (P1-s7) owing to the absence of macrovoids. Interestingly, the thickness of P1-s9 increases again to 88 μm, probably due to the stacking of growing spherulitic PVDF crystals. In line with the membrane structures, the porosity of the membranes decreases from 77% to 73% after 5 cycles of plasma treatment. Starting from 5 scan cycles, the membrane is dominated by the nodular structure (Fig. 2 (d4)–(d6)). The growth of polymer crystals results in a bigger empty space among the nodules, leading to a higher membrane porosity. Thus, the porosity of the PANIPS membranes increases to 81% when the scan cycle is further increased to 9.

As shown in Fig. 4f, the increased hydrophobicity and the elimination of macrovoids result in the PANIPS membranes with a higher LEP value when the scan cycle increases from 1 to 5. The maximum LEP of 2.43 bar is achieved by plasma scanning for 5 cycles. However, despite the similar hydrophobicity of P1-s5 ~ P1-s9, a longer treatment cycle results in a lower LEP because of the formation of the nodular structure (Fig. 2 (d4)–(d6)). As discussed in Supplementary Note 2 and Supplementary Fig. 6, the poor connection between the spherulitic PVDF crystals significantly reduces the mechanical properties. Thus, the membranes are more prone to liquid passage or cracking under high pressures before wetting occurs. Nevertheless, P1-s7 and P1-s9 membranes still have LEPs compared to the pristine NIPS membrane.

### Reveal the secrets of plasma treatments

Plasma, the fourth state of matters, is a partially ionized gas that comprises electrons, ions, metastable gas molecules, and neutral gas molecules. The mechanisms to improve membrane superhydrophobicity and alter the crystalline phase by plasma treatment will be studied and discussed in this section.

**Effects of plasma treatment on membrane superhydrophobicity.** To verify the underlying mechanisms of plasma treatment, the components in plasma can be categorized into three major parts for simplicity: (1) the high energy species (electrons, ions, and metastable species), (2) the electric field (formed by the voltage applied to discharge the gas), and (3) Ar gas blowing (neutral gas molecules)[44]. The effect of temperature increment is negligible, as the plasma treatment

for 9 cycles only increases the membrane temperature by 1 °C (Supplementary Fig. 7).

When the plasma is ignited, the high energy species endows plasma high reactivity that can interact with surrounding air and moisture, generating reactive oxygen and nitrogen species (RONS), such as OH, O, H, N, $NO_x$[45,46]. Previously, Jiang et al. used molecular beam mass spectroscopy to study the ionic species in Ar atmospheric pressure plasma and found that water ion clusters ($M(H_2O)_n$, where $M = O_2^-$, $CO_3^-$, $O_3^-$, $NO^-$, $NO_2^-$, $NO_3^-$, $HCO_3^-$, $H_3O^+$, $NO^+$, or $NO_2^+$) were the most dominant and stable ions after cascade reactions of RONS and the adjacent gases in the plasma effluent[47]. Another study by Gaens and Bogaerts also suggested the positive ions in plasma immediately clustered with water as soon as the air concentration in the plasma jet started to increase. The water clusters quickly became the most important charge carriers in the plasma and their sizes grew as a function of distance from the exit of a nozzle[48].

In PANIPS, it is speculated that the charged water ion clusters are generated in plasma and impinge on the membrane at a faster rate because of the acceleration by the electric field. The bombardment of water ion clusters on the membrane surface not only creates hierarchical micro/nano structures but also induces the solid-liquid demixing at the membrane surface, thereby rendering the membrane with superhydrophobicity. To verify our hypothesis, the weight increment was studied as a function of plasma scan cycle at various working distances. For comparison, the as-cast membranes without plasma treatment were also exposed to the same humidity environment (RH 70 ± 5%) to prepare VIPS membranes. As depicted in Fig. 5a, the weight of the VIPS membranes increases linearly with the exposure time in humid air, implying a slow absorption of water vapor from the atmosphere. In contrast, membranes treated with plasma at different distances have higher levels of weight increment, revealing that plasma indeed effectively facilitates water deposition on the membrane

surface. When increasing the working distance, the rate of weight increment slightly decreases, yet still higher than that of the VIPS membranes. The negative correlation between the rate of weight increment and the impinging distance supports the hypothesis of the electric field acceleration because it weakens at a longer distance. Additionally, gravity and Brownian diffusion may also facilitate the water deposition[49].

It is known that a strong air flow spraying on membrane surface can cause local distortion of the membrane surface and increase the roughness (i.e., the SANIPS method[36]). However, it is unlikely that the Ar gas blowing in the PANIPS method plays a dominant role in distorting the local membrane surface because of its small flow rate at only 100 sccm. Figure 5b shows the evidence that the membranes treated with either a flowing Ar gas without the electric field (i.e., no plasma discharge) or an electric field without Ar gas (i.e., no plasma) cannot change WCA and SA much. They have similar hydrophobicity to the NIPS membrane. Clearly, the formation of plasma and water clusters is the key to rendering the membranes with superhydrophobicity (Fig. 5b, P1-s9). While the Ar gas flow alone may not exhibit direct effectiveness, it likely aids convection, facilitating the transportation of water clusters to the membrane surfaces.

Because the formation of water ion clusters relies on the cascade reactions between the RONS in plasma and the moisture in air, the effect of relative humidity in atmosphere on membrane superhydrophobicity is further studied. As shown in Fig. 5b, the WCA of the membranes treated by plasma decreases from 162° to 107° when the relative humidity reduces from 70% to 30%, respectively. Meanwhile, the SA increases from 7° to 90° (non-slippery), suggesting the plasma treatment only works at sufficiently high humidity environments. To understand the underlying reasons, optical emission spectroscopy (OES) was applied to investigate the plasma composition at different relative humidity. As revealed in Fig. 5c, the transition lines of Ar 4p-4s

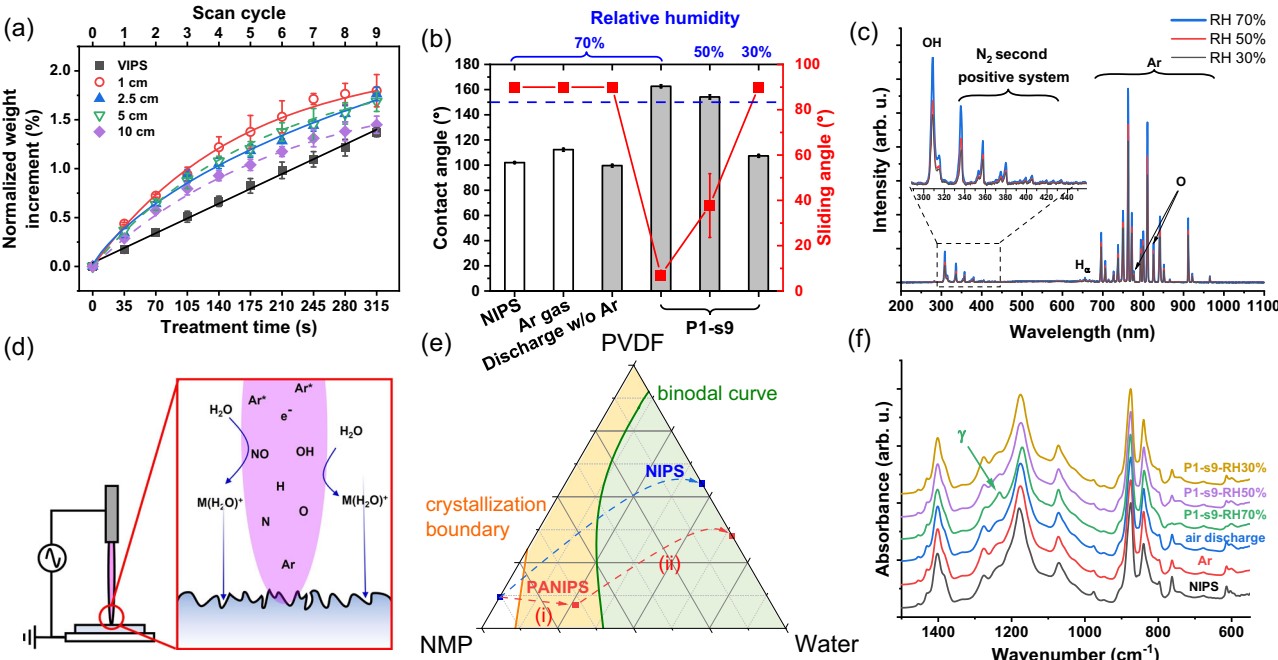

**Fig. 5 | Investigation of the underlying mechanisms of the PANIPS method.**
**a** The normalized weight increments of the VIPS and PANIPS membranes at different working distances from 1 cm to 10 cm. The temperature and humidity were controlled at 25 °C and RH 70%. Data were presented as the mean of 3 replicates ± standard deviation. **b** The WCA and SA of NIPS, Ar gas, discharge w/o Ar, and PANIPS membranes (P1-s9) prepared at different humidity from 70% to 30%. Data were presented as the mean of 4 replicates ± standard deviation. **c** The optical emission spectra of Ar plasma operated at different RH from 30% to 70%. **d** An

illustration of membrane formation mechanism during PANIPS. **e** A schematic phase diagram of the PVDF/NMP/water ternary system. Solid-liquid demixing (crystallization) occurs in both the orange and the green regions, while liquid-liquid demixing only occurs in the green region. The NIPS method (blue line) quickly brings the composition across the binodal curve, while the PANIPS method (red line) contains two steps: (i) plasma treatment and (ii) phase inversion in the coagulant bath. **f** The ATR-FTIR spectra of NIPS and PANIPS membranes prepared at different humidity.

locating in the range of 690-1000 nm are the primary components in Ar plasma. Additional components are the lines of RONS including OH band (309 nm), $N_2$ species (310-440 nm), and atomic oxygen (777.4 and 844 nm), that would react with surrounding humidity and form water ion clusters. A significant decline in the RONS lines (Fig. 5c, inset figure) is found when reducing the relative humidity from 70% to 30%, implying the less water ion clusters could be formed due to the lack of the reactants between RONS and moisture. As a result, the membrane treated with plasma at a low humidity of 30% is neither super-hydrophobicity nor slippery.

The above results allow to clarify the mechanisms of the PANIPS method (Fig. 5d). The reactive species in plasma first react with the moisture in the surrounding air, forming various water ion clusters. These charged water ion clusters are further deposited on the membrane surface due to the electric field acceleration, gravity, convection, or Brownian diffusion, creating hierarchical nanostructures beneficial for the superhydrophobicity. Moreover, the water clusters at the same time induce rapid phase separation and fix the coarsened surface. Comparing to the VIPS method, which requires long exposure time to achieve membranes with high hydrophobicity or near superhydrophobicity[32–34], the PANIPS method offers a more efficient and straightforward approach to fabricate superhydrophobic and self-cleaning membranes.

**Effects of plasma treatment on membrane crystalline phases**. Figure 5e shows different phase separation paths of NIPS and PANIPS in the PVDF/NMP/water ternary phase diagram. In NIPS membranes, the fast solvent and nonsolvent exchange brings the nascent membrane across the binodal curve, therefore the membrane precipitates through liquid-liquid demixing. In contrast, the composition change of the nascent membrane upon plasma treatment is slow, where water from various charged water ion clusters gradually brings the membrane composition into the crystallization region. This allows the formation and growth of the crystalline nuclei. Generally, the $\alpha$-phase has a molecular structure referred to as trans-gauche-trans-gauche (TGTG) and is thermodynamically stable, making it the preferred phase during crystallization. Conversely, the $\gamma$- and $\beta$-phases have conformations as TTTGTTTG and TTTT, respectively, and exist in a metastable state with a higher energy barrier during crystallization[40]. Therefore, $\alpha$-phase is thermodynamically favored over $\gamma$- and $\beta$-phases unless providing extra driving forces. It is known that the crystalline phases of PVDF can be influenced by several factors, such as the solvent power[50], solvent polarity[51], and solvent removal rate[52] during phase inversion. In addition, the phase transition can be achieved by blending PVDF with heterogeneous polymers or inorganic fillers in dope solutions to form local field-dipole interactions[53], or by electrical poling in the AC or DC electric field at a high temperature[54,55], as tabulated in Supplementary Table 2.

In the case of the PANIPS method, because the same solvent (i.e., NMP) was employed to prepare dope solutions for various membranes, the solvent effects on the transition of crystalline phases is neglected. On the other hand, the plasma does not introduce additional heterogeneous materials except various charged water ion clusters and the electric field. Thus, the difference in the phase composition could be ascribed to (1) the strong electrostatic interactions between the charged water ion clusters and the polar $\gamma$-phase[56], and/or (2) the electrical poling by the AC electric field in plasma. To clarify the mechanism, the IR spectra of the membranes treated (1) without discharge (i.e., only Ar gas), (2) discharge w/o Ar (i.e., only electric field), and (3) with plasma (P1-s9 in different humidity) are analyzed. As shown in Fig. 5f, only the P1-s9 membrane treated at RH 70% shows the distinct $\alpha \rightarrow \gamma$ transition. In other words, the electrical poling by the plasma is not the main driving force that facilitates the phase transition.

To further verify if the enhanced water intake is the only reason to cause the phase transition, a PANIPS membrane was fabricated and compared with three control membranes prepared using the NIPS, VIPS, and SANIPS methods under identical humidity, temperature, and treatment time (~5 min, the same as PANIPS for 9 cycles). The VIPS and SANIPS methods were used for comparison because they promote the water intake in different degrees without the electric field (Supplementary Note 3). As shown in Supplementary Fig. 8a, the amount of water intake during the early membrane formation follows the order of PANIPS ≈ SANIPS > VIPS when the PANIPS working distance is at 2.5 cm ~ 5 cm. When comparing the IR spectra among the P2.5-s9 membrane, SANIPS and VIPS membranes, it is apparent that only the P2.5-s9 membrane shows a fully converted $\gamma$-phase spectrum, as illustrated in Supplementary Fig. 8b. In contrast, the SANIPS membrane still exhibits a $\alpha$-phase characteristic peak at 763 cm$^{-1}$ and a slightly increased $\gamma$-phase characteristic peak at 1234 cm$^{-1}$. The VIPS membrane is similar to the NIPS membrane, both having strong $\alpha$- and $\beta$-phase characteristic peaks. This implies that although water plays an important role in inducing the $\alpha \rightarrow \gamma$ phase transition, the electric field in plasma (~tens of kV cm$^{-1}$[57]) might be the additional driving force that facilitates the phase transition in the PANIPS method. Furthermore, it should be noted that the water ion clusters deposited on the membranes would have a low pH because of the acidic ions (i.e., $CO_3^-$, $NO^-$, $NO_2^-$, $NO_3^-$, $HCO_3^-$, $H_3O^+$, $NO^+$, or $NO_2^+$) that coupled with water clusters[47]. The impact of the low pH nonsolvent on membrane crystallinity remains unclear and requires further investigations in the future.

## Membrane performance
**Piezoelectric performance of PANIPS membranes**. Because the plasma treatment significantly promotes the electroactive phases ($\gamma$ and $\beta$) in the PANIPS membranes, a homemade test platform was used to assess the improved piezoelectric property. The generation of piezoelectric voltage involves two steps: (1) When being pressurized, the membrane deformation creates a potential difference between the upper and the lower electrodes, producing an output voltage; (2) Releasing the pressure produces a reversed voltage due to the reversal of polarization. As shown in Fig. 6a, the pristine NIPS membrane has a weak output voltage of 0.73 $V_{pp}$ (peak-to-peak voltage) because of the existence of ~30% non-electroactive $\alpha$-phase (Supplementary Fig. 4) in the membrane. Comparing to NIPS, P1-s1 and P1-s3 show lower output voltages of 0.17 $V_{pp}$ and 0.24 $V_{pp}$, respectively (Fig. 6b–c). This can be ascribed to the less compressibility due to the elimination of macro-voids. With increasing scan cycles to 5, 7, and 9, the output voltages increase drastically to 6.89 $V_{pp}$, >10 $V_{pp}$, and 7.34 $V_{pp}$ (Figs. 6d–f). The changes in piezoelectric output voltage are in accordance with the ratio of electroactive phases and the total crystallinity. As shown in Fig. 4d and Supplementary Fig. 4, the high content (~40%) of $\alpha$-phase and the low crystallinity of P1-s1 and P1-s3 result in their lower output voltages than the NIPS membrane. Starting from 5 scans, the $\alpha$-phase is fully converted to electroactive $\gamma$ and $\beta$ phases, and the crystallinity also increases at more scans. As a result, the output voltages of P1-s5 ~ P1-s9 are all much higher than the NIPS membrane. The piezoelectric constant d33 was also measured to confirm the piezoelectricity of the PANIPS membranes. As depicted in Supplementary Fig. 9, the d33 value of the pristine NIPS membrane is 0.2 pC N$^{-1}$, and it continuously increases as a function of the plasma scan cycle. When being treated by plasma for 5, 7, and 9 cycles, the d33 increases significantly to 8.9, 9.7, and 10.5 pC N$^{-1}$, respectively. Interestingly, despite P1-s9 having the highest d33 value, its output voltage is lower than that of P1-s7. This discrepancy might stem from the fact that although the d33 measurement is geometry-independent, the output voltage can be influenced by various factors such as compression area, membrane thickness, reciprocating frequency, and other piezoelectric coefficients[58,59]. Therefore, a further in-depth study is required to assess the underlying reasons in the future.

Overall, the results verify PANIPS as a facile and effective approach to promote piezoelectric performance of PVDF membranes.

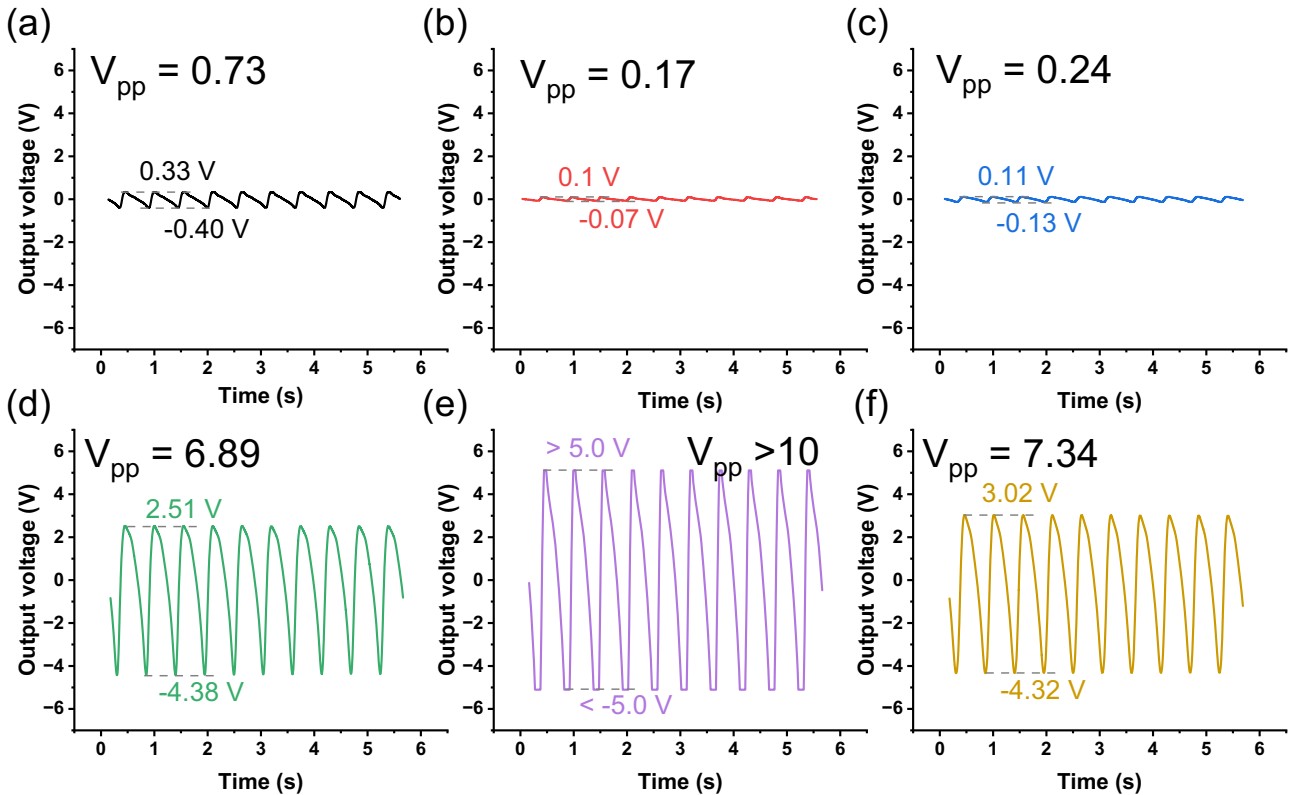

**Fig. 6 | Waveforms of piezoelectric voltage output of NIPS and PANIPS membranes. a** NIPS, **b** P1-s1, **c** P1-s3, **d** P1-s5, **e** P1-s7, and **f** P1-s9. The positive and negative peaks were the piezoelectric voltages when the membrane was pressurized and released, respectively. The membrane with an area of 1.5×1.5 cm2 was tested by pressurization/release cycles with 1 N (-4.4 kPa) and the frequency of 1.82 Hz.

Traditionally, there are two common methods to improve the piezoelectric properties of PVDF membranes: electrical poling and localized poling with additives[60,61]. Electrical poling is time-consuming (often taking several hours) and can be hazardous due to the high voltage needed to create a strong electric field (ranging from MV/m to GV/m)[62]. On the other hand, localized poling with additives increases overall costs. In contrast, the PANIPS method offers a quicker process without the need for additives to achieve highly piezoelectric membranes. As indicated in Supplementary Table 2, it's noteworthy that the output voltage and the d33 value of PANIPS membranes are comparable to, or in some cases even surpass, those of composite or electrically poled PVDF membranes. This suggests the PANIPS is a promising method to yield membranes with excellent piezoelectric properties.

**DCMD performance.** Similar to other membrane separation processes, traditional MD membranes encounter issues such as scaling, fouling, and wetting. Enhancing the hydrophobicity of the membrane is an effective strategy to relieve these problems. As mentioned in the previous discussion, the plasma treatment imparts superhydrophobicity and self-cleaning properties to the PANIPS membranes. Therefore, it is expected the PANIPS membrane would exhibit better wetting and fouling resistance. For verification, several feed solutions including (1) a high salinity brine solution (10 wt% NaCl), (2) brine solutions containing foulants (Rose Bengal dye or humic acid), and (3) brine solutions containing the surfactant (SDS) were prepared for DCMD tests. The NIPS membrane serves as a benchmark for comparison with the PANIPS membrane. The P1-s5 membrane was selected due to its highest LEP and superhydrophobic properties among all PANISP membranes.

When treating a 10 wt% NaCl feed solution, both the NIPS and PANIPS membranes exhibit a stable vapor flux and rejection over the 24 hr tests (Fig. 7a). It is worth noting that the PANIPS membrane shows a significantly higher flux compared to the NIPS membrane (20.7 vs. 9.6 kg m⁻² hr⁻¹), which could be attributed to the increased surface roughness and porosity resulting from the plasma treatment.

The improved fouling resistance of the PANIPS membranes is further verified by introducing foulants into the 10 wt% NaCl feed solution. Two model foulants, Rose Bengal dye and humic acid, were selected to represent common organic dyes and natural organic matter (NOM) compounds that often lead to fouling during the MD process[12,63]. As depicted in Fig. 7b–c, the vapor flux of the PANIPS membrane remains stable over the 10 hr test, whether treating the Rose Bengal dye solution or the humic acid solution. In contrast, the fluxes of the NIPS membrane in both cases decrease about 40%, indicating the occurrence of fouling and pore blocking by the dye molecules and salt crystals. The DCMD tests confirm that the slippery and superhydrophobic PANIPS membranes can effectively alleviate the membrane from fouling. The same fouling resistance is also validated using other PANIPS membrane (P1-s9) to treat the same brine and dye solutions, as discussed in Supplementary Note 4 and Supplementary Fig. 10.

To compare the wetting resistance between the NIPS and PANIPS membranes, SDS is progressively added to the 10 wt% NaCl solution every 2 hr (up to 0.2 mM) to lower the surface tension and accelerate the wetting process (Fig. 7d)[64]. For the NIPS membrane, partial-wetting takes place when the SDS concentration reaches 0.1 mM, as indicated by the decreasing flux and rejection. Further increasing the SDS concentration to 0.2 mM results in full wetting, where the flux drastically increases, accompanied with a sharp decrease in the rejection. In contrast, the performance of the PANIPS membrane is more robust, showing that the enhanced superhydrophobicity can mitigate the surfactant-induced pore wetting[65].

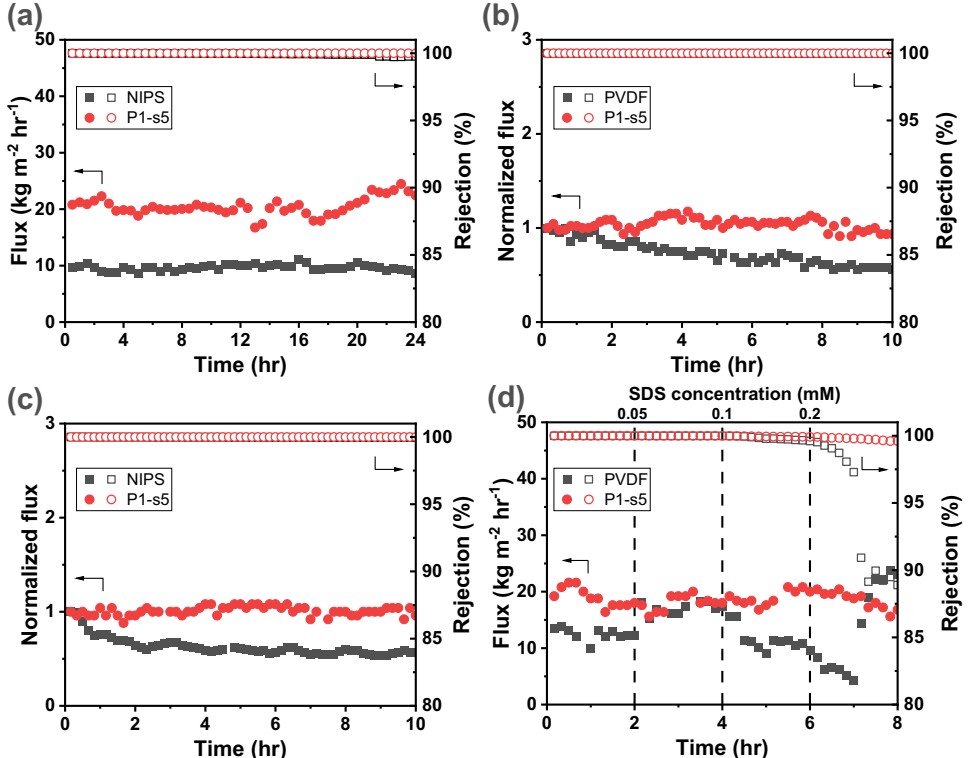

**Fig. 7 | DCMD tests of NIPS and P1-s5 membranes with different feed solutions.** **a** 10 wt% NaCl; **b** 1000 ppm Rose Bengal dye in 10 wt% NaCl, where the flux values of NIPS and P1-s5 were normalized by their initial values of 12.3 kg m$^{-2}$ hr$^{-1}$ and 16.2 kg m$^{-2}$ hr$^{-1}$, respectively; **c** 500 ppm humic acid in 10 wt% NaCl, with the flux values of NIPS and P1-s5 normalized by their initial values of 12.6 kg m$^{-2}$ hr$^{-1}$ and 15.0 kg m$^{-2}$ hr$^{-1}$, respectively; **d** Different SDS concentrations in 10 wt% NaCl,

where SDS was incrementally added to the feed tank every 2 hours until the SDS concentration reached 0.2 mM. All MD tests were conducted with feed solutions maintained at 60 °C, and initially, DI-water was used as the permeate and controlled at 15 °C. The solid and hollow symbols represent the vapor flux and rejection, respectively.

Finally, the long-term performance of the PANIPS membrane was conducted to prove its potential in real applications. A solution containing 100 ppm Rose Bengal and 5 wt% NaCl was used as the feed. This composition was chosen because it was closed to that in real textile wastewater[63]. As displayed in Supplementary Fig. 11, the flux remains unchanged and the rejections are all higher than 99.9% throughout the continuous 100 hr test, implying the great stability of the PANIPS membrane.

## Discussion

This work presents a facile PANIPS method for preparing superhydrophobic, self-cleaning, and piezoelectric PVDF membranes. It discovered that imposing plasma on as-cast membranes creates multilevel roughness, increases surface porosity, and suppresses the formation of macrovoids. The enhanced surface roughness and membrane porosity render the membranes with superhydrophobic and self-cleaning characteristics. In addition, plasma treatment promptly induces a unique crystalline phase transition from nonelectroactive α-phase to electroactive γ-phase. The morphological and physicochemical properties of PVDF membranes can be adjusted by carefully controlling the plasma treatment parameters, such as scan cycle, scan speed, and working distance. Without any additive and post-treatment, the PVDF membranes prepared by the PANIPS method exhibit remarkable piezoelectric performance with the maximum output voltage of >10 V$_{pp}$. In addition, they demonstrate excellent resistance to various aqueous solutions and exhibit a higher flux, greater salt rejection, and superior long-term stability to NIPS membranes when treating feed solutions containing 10 wt% NaCl and Rose Bengal via DCMD. The mechanisms of the PANIPS method have also been systematically studied and proposed. It is believed that the water

clusters and electric field in the plasma synergistically enhance the superhydrophobicity and induce the transition of the crystalline phase.

On the other hand, the study is still in its early stage because this work only investigates the PANIPS method using a simple Ar microplasma. Further studies should be conducted to investigate the effects of nozzle size, plasma frequency, and applied voltage. Since plasma is a versatile tool widely used in various applications, including the deposition of polymeric thin films and the synthesis of nanomaterials in solutions. Expanding the PANIPS method to incorporate different reaction gases or adding precursors to the dope solutions may open up totally new possibilities for tailoring the physicochemical properties of membranes.

It is worth highlighting that the PANIPS method is carried out in atmosphere, making it more practical for scale-up production since no vacuum system is involved. Its simplicity, green nature, and ability to produce superhydrophobic, self-cleaning and piezoelectric membranes with exceptional properties make it a compelling alternative to traditional methods. With more research in the future, the PANIPS method holds significant promise in the development of high-performance functional membranes for diverse applications.

## Methods

### Materials and chemical reagents

Polyvinylidene difluoride (PVDF, KYNAR® HSV900) was received from Arkema. Ethylene glycol (EG), N-Methyl-2-pyrrolidinone (NMP, >99%), humic acid (HA, >90%), and sodium dodecyl sulfate (SDS) was ordered from Sigma-Aldrich. Sodium chloride (NaCl) was received from DUKSAN. All chemicals were used as received without further purification.

**Table 1 | The parameters used for microplasma treatments**

| microplasma treatment parameters | |
|---|---|
| applied voltage (kV) | 7 |
| frequency (kHz) | 20 |
| gas flow rate (sccm) | 100 |
| working distance (cm) | 1, 2.5, 5, 10 |
| x scan rate (mm s⁻¹)/step (mm) | 20/10 |
| y scan rate (mm s⁻¹)/step (mm) | 50, 100, 150, 200, 250/100 |
| scan area for each cycle (mm) | 120 × 100 |
| scan cycle | 1, 3, 5, 7, 9 |

## Membrane preparations via the PANIPS method

The experimental procedures were depicted in Fig. 1. The dope solution was prepared by dissolving PVDF, EG, and NMP with a weight ratio of 12/10/78, followed by constantly stirring at 60 °C for 12 hr. Afterwards, the dope solution was naturally cooled down to room temperature and cast on a glass plate using a casting knife with a gap height of 300 μm. PANIPS membranes were fabricated by treating the as-cast membrane with Ar microplasma. During the microplasma treatment, the as-cast membrane was placed on the stage that was programmed to move with a step of 10 and 100 mm along x- and y-axis, respectively, to scan an area of $12{\times}10$ cm² for different cycles. Different treatment parameters including the effects of working distance (i.e., the needle-to-membrane distance), scan cycle, and scan speed were investigated and summarized in Table 1. The total time required for 1 scan cycle with an area of $10{\times}12$ cm² was approximately 35 s, given the conditions of the moving speed on the x and y-axes at 20 mm s⁻¹ and 50 mm s⁻¹, respectively. After the plasma treatments, the membranes were immediately immersed in a tap-water coagulant bath to complete the phase inversion for 20 hr, followed by freeze drying (Alpha 2-4 LSCplus, Martin Christ) to preserve their pores. The resultant membranes were named as P$m$-s$n$, where $m$ indicated the working distance and $n$ represented the scan cycle. Control PVDF membranes were prepared by the NIPS method following the same procedures without plasma treatment as described previously. All experiments were conducted at room temperature of 20 ~ 25 °C and a relative humidity (RH) of 65 ~ 70% if not further specified. The nascent membrane temperature during plasma treatment was measured using an infrared thermal imager (Fluke Ti450, Comark Instruments, USA).

## Characterizations

**Surface morphology.** A field emission scanning electron microscope (FE-SEM, 6500F, JEOL) was applied to observe the top surface and the cross-sectional morphology of PANIPS membranes. The membrane samples were cracked in liquid nitrogen to observe their cross-sectional structures. The surface topography was accessed by atomic force microscopy (AFM, Dimension ICON, Bruker) with a scan size of $25 \times 25$ μm². To monitor the formation of the skin layer after microplasma treatment, images of nascent membranes before transferring to the coagulant bath were taken by a phase-contrast microscope.

**Crystalline polymorphisms of membranes.** The surface chemical functionalities and the crystalline phases of PVDF were examined by attenuated total reflectance Fourier transformed infrared spectroscopy (ATR-FTIR, Spectrum One, PerkinElmer) in the wavenumber ranging from 4000 cm⁻¹ to 500 cm⁻¹, with a resolution of 1 cm⁻¹ and a scan number of 16. The intensities of the characteristic peaks at 763 cm⁻¹ (α phase), 1234 cm⁻¹ (γ phase), and 1275 cm⁻¹ (β phase) were used to identify the changes of the PVDF crystalline phases[40]. The fraction of electroactive phase (F$_{EA}$, including β and γ phases) was

quantitatively determined using Eq. 1) proposed by Cai et al.[42]

$$F_{EA}(\%) = \frac{I_{EA}}{\left(\frac{K_{840^*}}{K_{763}}\right)I_{763} + I_{EA}} \times 100\% \tag{1}$$

where $I_{EA}$ and $I_{763}$ are the absorbencies at 840 cm⁻¹ and 763 cm⁻¹, respectively; $K_{840^*}$ and $K_{763}$ are the absorption coefficients at the respective wave numbers with the values of $7.7 \times 10^4$ and $6.1 \times 10^4$ cm² mol⁻¹, respectively. The crystalline phases of the PVDF membranes were also confirmed by the X-ray diffraction (XRD, Bruker D2 PHASER XE-T XRD). The total crystallinity (Xc, %) of PANIPS membranes was analyzed by the differential scanning calorimeter (DSC25, TA Instruments) in a nitrogen atmosphere. During experiments, the temperature was elevated from 40 to 200 °C at a heating rate of 10 °C min⁻¹ to measure the enthalpy change (ΔH$_f$). By dividing the measured ΔH$_f$ of PVDF membranes to that of the perfect PVDF crystals (ΔH$_f^o$ = 105 J g⁻¹), Xc can be determined using Eq. (2)[66]:

$$X_c(\%) = \frac{\Delta H_f}{\Delta H_f^0} \times 100\% \tag{2}$$

**Membrane wettability and self-cleaning properties.** The surface wettability of PANIPS membranes was measured by a goniometer (OCA25, DataPhysics Instruments) using the sessile drop method. At least four droplets (6 uL) were measured to acquire the averaged WCA. The same goniometer was applied to access the self-cleaning ability of the PANIPS membranes by measuring their SAs. In SA measurements, a water droplet of 10 μL was placed on the membrane and the stage was tilted at the speed of 1°/s from 0° to 90°. The tilted angle at which the droplet started to slide was recorded as the SA.

**Liquid entry pressure.** The LEPs of membranes were determined using a custom-designed cell adopted from[64]. In brief, the cell consisted of two chambers, separated by the membrane being tested and a porous support. To prevent leakage, the membrane (with an effective diameter of 1.5 cm) was secured onto the porous support and sealed using an O-ring. Prior to each measurement, the load chamber was filled with a 10 wt% NaCl solution, and the entire cell was immersed in a 250 mL beaker filled with DI-water that was continuously stirred at 400 rpm. The load chamber was then pressurized with N₂ gas. Starting from 0.8 bar, the pressure was increased with an increment of 0.2 bar every 2 min until the LEP was reached. The conductivity of DI-water was monitored using a conductivity meter during the measurements. The LEP was defined as the pressure at which the conductivity of DI-water increased abruptly due to the passage of the saline solution in the load chamber across the membrane. Three measurements were taken to obtain the averaged LEP for each membrane.

**Membrane pore size and porosity.** The mean pore size was characterized by a capillary flow porometer (CFP-1500AE, Porous Materials Inc) via the wet-up/dry-down method using Galwick and nitrogen as the wetting solution and drying gas, respectively. The surface porosity was calculated by analyzing the surface FE-SEM images by the ImageJ software. The overall porosity was calculated using Eq. (3)[37]:

$$Porosity(\%) = \left(1 - \frac{m_p}{\rho_p A \sigma}\right) \times 100\% \tag{3}$$

where $m_p$ (kg), $\rho_p$ (kg/m³), A (m²), and σ (m) were the mass, polymer density (1.78 kg/m³), membrane area, and the averaged thickness, respectively.

**Moisture condensation tests.** The masses of the NIPS and PANIPS membranes were measured for 315 s (equal to the duration of plasma scan for 9 cycles) by a 4-digit balance (Shimadzu, ATX224R). A $1 \times 3$ cm²

membrane was cast on a glass slide by a casting knife with a gap height of 300 μm. The weight of the as-cast membrane on the glass slide was recorded as the initial value ($W_0$). Afterwards, the membrane was treated with plasma at different distances. The weights after being treated for 1, 3, 5, 7, and 9 cycles were measured ($W_S$) and subtracted by the initial value ($W_0$), and then normalized by the membrane area (A). The normalized weight increment was calculated using Eq. (4)[36]:

$$\Delta m(\%) = (W_s - W_0)/(W_0 - W_g) * 100\% \tag{4}$$

where $W_0$ (g) is the mass of the as-cast membrane on the glass slide; $W_S$ (g) is the weight of the membrane on the glass slide scanned by plasma for *n* cycles, where n = 1, 3, 5, 7, and 9; $W_g$ (g) is the mass of the glass slide.

**Tensile tests.** The mechanical properties of membranes were measured via a tensiometer (34SC-05, Instron). All measurements were conducted following the standard ASTM-D882. Specimens were prepared with the size of 15 mm in width and 100 mm in length. The starting gauge length, elongation speed, and gauge width were set as 50 mm, 10 mm/min, and 15 mm, respectively. At least 3 samples were tested for each membrane to obtain the averaged maximum tensile stress, maximum tensile strain, and Young's modulus.

## Membrane performance

**Piezoelectric performance tests.** To test the piezoelectric performance of a PANIPS membrane, both sides of the membrane were covered by copper tapes as electrodes, attached with thin silver-jacketed wires. The tests were carried out by a homemade reciprocating testing machine that could exert a periodic dynamic pressure at a fixed frequency of 1.82 Hz as shown in Supplementary Fig. 12[67,68]. Briefly, the prepared membrane was placed on the holder and the wire was connected to the signal processor that connected the device and the measurement circuit. A reciprocating pressure of 4.4 kPa (~1 N) accurately controlled by a linear motor was applied on the membrane with an area of $1.5 \times 1.5$ cm$^2$ to generate a piezoelectric voltage, which was then collected and processed by the signal processor unit.

The piezoelectricity coefficient d33 was directly measured using a wide range d33 tester meter (APC International, Ltd.)[61]. The membrane samples were metalized by the silver paste from both sides with an area of $1 \times 1$ cm$^2$, followed by covering both sides with copper foils. The metalized sample was clamped between two metallic jaws with a static force; the position of the upper jaw was static during the measurement, while the bottom jaw was excited by a harmonic mechanical oscillation force with an amplitude of 0.25 N and a frequency of 110 Hz. At least three replicates of each membrane were measured to obtain the averaged d33 value.

**Direct contact membrane distillation (DCMD) tests.** The DCMD tests were conducted using a lab-scale setup as shown in Supplementary Fig. 13. A membrane with an effective area of 2 cm$^2$ was mounted in a polymethylmethacrylate (PMMA) module holder with the microplasma treated surface facing to the feed solution. The feed solution was maintained at 60 °C and circulated at a flow rate of 0.15 L/min. On the other hand, the distillated tank was initially filled with de-ionized (DI) water and circulated at a flow rate of 0.1 L/min at 15 °C. Different feed solutions were applied to evaluate the wetting resistance and self-cleaning ability of the PANIS membranes during DCMD tests. A 10 wt% NaCl solution was utilized to demonstrate the membrane performance when treating high salinity brine solution. To further demonstrate the self-cleaning properties, two types of foulants (1000 ppm Rose Bengal dye, or 500 ppm humic acid) were added in to the 10 wt% NaCl solution for fouling resistance DCMD tests. Finally, the wetting resistance was studied by progressively adding SDS into 10 wt% NaCl every 2 hr to lower the surface tension of the feed solution during DCMD tests[64].

During MD experiments, the weight and conductivity of the distillate tank were measured continuously to acquire the real-time flux and salt rejection. The flux ($J_w$, kg/m$^2$ hr) was derived by Eq. (5):

$$J_w = \frac{\Delta W}{A \times \Delta t} \tag{5}$$

where $\Delta W$ (kg) is the weight change in the distillate over a duration of $\Delta t$ (h); A (m$^2$) is the effective membrane surface area contacting with the feed. The salt rejection R (%) was determined by Eq. (6):

$$R = \left(1 - \frac{C_d}{C_f}\right) \times 100\% \tag{6}$$

where $C_d$ and $C_f$ are the salt concentrations (M) of the distillate and feed, respectively. Considering the dilution effect, $C_d$ was determined by Eq. (7):

$$C_d = \frac{C_1 m_1 - C_0 m_0}{m_1 - m_0} \tag{7}$$

where $m_0$ and $m_1$ are the weights (kg) of the distillate at the initial and final stages, respectively, and $C_0$ and $C_1$ indicate the initial and final salt concentrations (M) of the distillate stream.

## Data availability
All data supporting the findings of this study are available within the article and the supplementary information. Any additional data are available from the corresponding author.

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

## Acknowledgements

The authors would like to thank the National Science and Technology Council (NSTC), Taiwan, for funding this research under Grant Award NO. NSTC 110-2222-E-011-022-MY3 (T.S.C.) and NSTC 111-2221-E-011-008-MY3 (M.-J.W.). Prof. Chung also thanks the Yushan Scholar Program supported by the Ministry of Education, Taiwan. Thanks are also due to Dr. Cong-Han Huang and Prof. Wei-Song Hung of NTUST for their precious suggestions on piezoelectric property analyses and Drs. K. J. Lu and Y. M. L. Chen of National University of Singapore for their help on the MD setup. Finally, the authors thank Arkema Inc. for the provision of PVDF in this study.

## Author contributions

Y.-H.H. designed the experiments, carried out the experiments, analyzed the data, and prepared the manuscript. M.-J.W. provided plasma equipment and analysis tools. T.S.C. supervised the study. T.S.C. and M.-J.W. provided constructive suggestions for the manuscript revision.

## Competing interests

The authors declare no competing interests.
