## [Peer Review File · Nature Communications]

Development of Multifunctional Membranes via Plasma-Assisted Nonsolvent Induced Phase SeparationREVIEWER COMMENTS

Reviewer #1 (Remarks to the Author):

This paper entitled with 'Novel Plasma-Assisted Non-Solvent Induced Phase Separation Method Towards Superhydrophobic, Self-Cleaning, and Piezoelectric PVDF Membranes' introduced a facile membrane fabrication method for membranes with high roughness. The authors manipulated the membrane morphology by adjusting the time of the micro-plasma treatment, and then tested membranes' wettability and self-cleaning property. The self-cleaning test was performed by immersing membranes in a 500 ppm Rose Bengal dye solution. The P1-s9 membrane, which was plasma scanned for 9 cycles, showed no residual solution, while pristine NIPS membrane became visibly pink. The authors didn't discuss the reason or the mechanism for this kind of self-cleaning.

And so is the DCMD process, the authors only give one comparison for the NIPS membrane and PANIPS membrane. It can not prove that its self-cleaning effect is very good with using the single foulant, and multiple foulant types will be a good choice. There is one contradictory in the Fig. 9(a). The NIPS membrane was wetted at about 3 hour, why the rejection decreased sharply at about 20 hour. It's unreasonable. Also, the authors didn't give the explanation for the anti-wetting property for the PANIPS membrane. The authors should clarify where the self-cleaning and anti-wetting performance come from, the super-hydrophobicity or the piezoelectricity.

As for the superhydrophobic property, the increasement of the roughness is vivid, presenting a remarkable increase in the water contact angle from about 95° to 166.4° and sliding angle. Please add XPS characterization to figure out if there is new functional group formed on the membrane surface.

For the waveforms of piezoelectric voltage output of NIPS and PANIPS membranes, where the output of the NIPS membrane come from? Since the NIPS membrane wasn't poled, it should not possess the piezoelectricity. Page 20, Line 450-451, "the changes in piezoelectric output voltage are in accordance with the ratio of the electroactive phases and the total crystallinity." This sentence is totally wrong. The piezoelectric output is mainly related to the C-F dipole arrangement within the membrane matrix. The piezoelectricity should be evidenced by d33 test. d33 is a vital parameter for the piezoelectric material.

Actually, the authors presented a new way for modifying the membrane fabrication and the work is heavy. However, the authors didn't organize the work very well. From my point of view, the presented self-cleaning property is caused by the rough surface, which is not related to the piezoelectricity. And for the self-cleaning and piezoelectricity, the authors didn't discuss in depth for the origin and the relevance.

Thus, this manuscript does not reach the level of Nature Communications.

Reviewer #2 (Remarks to the Author):

The paper describes a new technique for membrane manufacturing that exploits an Ar microplasma to promote the phase separation process, enhancing the membrane surface roughness and favours the formation of piezoelectric PVDF phases.

The manuscript describes thoroughly this new technique, showing the effects of several preparation parameters of the structure and performance of the membranes. This is clearly a high value research that propose a very innovative technique. Therefore, I think that the paper can surely be published on nature communications journal.

However, out of pure curiosity, I would like the authors to answer some questions:

How long did the plasma take to treat the whole sample?

After the treatment the sample was immediately immersed in water, or some time was required to do this operation?

How difficult would it be to scale this setup to an industrial production.

On page 9 the authors state "... LEP because of the formation of the nodular structure and the weak mechanical properties of the spherulitic PVDF crystals." I think that the problem could be the fragility of the connections between the spherulites more than the mechanical properties of the spherulites themselves. Did the authors evaluate the tensile strength at break of the membranes?

On page 15, the Ar flow rate is expressed in sccm. I suggest to use a clearer nomenclature such as cmstd/min.

Reviewer #3 (Remarks to the Author):

This manuscript reports discovery of a new approach for PVDF membrane fabrication based on plasma-assisted non-solvent induced phase separation. The resulting membrane is superhydrophobic, and is found to be quite effective when applied for direct contact membrane distillation (DCMD). The study is innovative and systematic, focused on illustrating the underlined membrane formation mechanisms. The manuscript is well prepared and easily to follow. Listed below are some minor comments that should be considered during the revision process.

- 1) Abstract: There appears to have no need to mention motion sensor, CO2 capture etc.; those are not discussed in the main body of the paper.
- 2) P7 and figure 2: What is the duration for each cycle of atmospheric pressure plasma treatment? Plasm temperature? The resolution of SEM images appears low.
- 3) Figure 6: Water clusters are considered important for the membrane development process, but there appears to be no direct evidence regarding the cluster population and sizes.
- 4) P16: The statement "The charged water ion clusters are then accelerated by the electric field ..." is

questionable. AC, not DC, is used for the plasma production, so the ions are not moving towards a specific direction under the electric field. Brownian motion could lead to attachment of water clusters to the dope liquid surface.

5) P20/Figure 8: Piezoelectric phenomena for PVDF is not new. Is the V_{pp} observed there the highest when compared with PVDF prepared by other approaches?

List of Changes in the Manuscript:

Manuscript Title: A Novel Plasma-Assisted Non-Solvent Induced Phase Separation Method Towards Superhydrophobic, Self-Cleaning, and Piezoelectric PVDF Membranes

Manuscript ID: NCOMMS-23-36810

We thank the reviewers' valuable comments to improve the quality of the manuscript. The manuscript has been revised thoroughly according to each reviewer's comments. Below please find our point-by-point responses to each comment.

Reviewer's comments:**Reviewer #1 (Remarks to the Author):**

This paper entitled with 'Novel Plasma-Assisted Non-Solvent Induced Phase Separation Method Towards Superhydrophobic, Self-Cleaning, and Piezoelectric PVDF Membranes' introduced a facile membrane fabrication method for membranes with high roughness. The authors manipulated the membrane morphology by adjusting the time of the micro-plasma treatment, and then tested membranes' wettability and self-cleaning property.

Response: We thank the reviewer for taking time to review our manuscript and provide constructive comments to improve the manuscript. The following lists our point-to-point replies.

Comment 1. The self-cleaning test was performed by immersing membranes in a 500 ppm Rose Bengal dye solution. The P1-s9 membrane, which was plasma scanned for 9 cycles, showed no residual solution, while the pristine NIPS membrane became visibly pink. The authors didn't discuss the reason or the mechanism for this kind of self-cleaning.

Reply: In lines 183-193 on page 9, we have elaborated on how the plasma treatment can substantially enhance surface roughness and porosity in the membranes, resulting in super-hydrophobicity and a significantly reduced sliding angle. This augmentation in surface characteristics (i.e., super-hydrophobicity and low sliding angle) is the key factor driving the self-cleaning capability of the membranes. We have added a clearer explanation after self-cleaning tests as the reviewer suggested as follows.

[Page 9, lines 205-208] ... *The improved self-cleaning characteristic of PANIPS membranes can be primarily attributed to their increased surface roughness and porosity, which function similarly to the lotus effect in maintaining the cleanliness of*

the membrane surface.

Comment 2. And so is the DCMD process, the authors only give one comparison for the NIPS membrane and PANIPS membrane. It can not prove that its self-cleaning effect is very good with using the single foulant, and multiple foulant types will be a good choice.

Reply: We thank the reviewer for the suggestion. Additional fouling and wetting tests were conducted using humic acid and a common surfactant (i.e., sodium dodecyl sulfate (SDS)). Both results confirm that the PANIPS membranes have better durability when treating versatile organic foulants or surfactants. We have added more discussion in the DCMD part on pages 23-24 as follows.

[pages 23-24, lines 502-539] *Similar to other membrane separation processes, traditional MD membranes encounter issues such as scaling, fouling, and wetting. Enhancing the hydrophobicity of the membrane is an effective strategy to relieve these problems. As mentioned in the previous discussion, the plasma treatment imparts superhydrophobicity and self-cleaning properties to the PANIPS membranes. Therefore, it is expected the PANIPS membrane would exhibit better wetting and fouling resistance. For verification, several feed solutions including (1) a high salinity brine solution (10 wt% NaCl), (2) brine solutions containing foulants (Rose Bengal dye or humic acid), and (3) brine solutions containing the surfactant (SDS) were prepared for DCMD tests. The NIPS membrane serves as a benchmark for comparison with the PANIPS membrane. The P1-s5 membrane was selected due to its highest LEP and superhydrophobic properties among all PANISP membranes.*

When treating a 10 wt% NaCl feed solution, both the NIPS and PANIPS membranes exhibit a stable vapor flux and rejection over the 24 hr tests (Fig. 7a). It is worth noting that the PANIPS membrane shows a significantly higher flux compared to the NIPS membrane (20.7 kg/m² hr vs. 9.6 kg/m² hr), which could be attributed to the increased surface roughness and porosity resulting from the plasma treatment.

The improved fouling resistance of the PANIPS membranes is further verified by introducing foulants into the 10 wt% NaCl feed solution. Two model foulants, Rose Bengal dye and humic acid, were selected to represent common organic dyes and natural organic matter (NOM) compounds that often lead to fouling during the MD process^{12, 60}. As depicted in Fig. 7b-7c, the vapor flux of the PANIPS membrane remains stable over the 10 hr test, whether treating the Rose Bengal dye solution or the humic acid solution. In contrast, the fluxes of the NIPS membrane in both cases decrease about

40%, indicating the occurrence of fouling and pore blocking by the dye molecules and salt crystals. The DCMD tests confirm that the slippery and superhydrophobic PANIPS membranes can effectively alleviate the membrane from fouling. The same fouling resistance is also validated using other PANIPS membrane (P1-s9) to treat the same brine and dye solutions (Fig. S12).

To compare the wetting resistance between the NIPS and PANIPS membranes, SDS is progressively added to the 10 wt% NaCl solution every 2 hr (up to 0.2 mM) to lower the surface tension and accelerate the wetting process (Fig. 7d).⁶¹ For the NIPS membrane, partial-wetting takes place when the SDS concentration reaches 0.1 mM, as indicated by the decreasing flux and rejection. Further increasing the SDS concentration to 0.2 mM results in full wetting, where the flux drastically increases, accompanied with a sharp decrease in the rejection. In contrast, the performance of the PANIPS membrane is more robust, showing that the enhanced superhydrophobicity can mitigate the surfactant-induced pore wetting.⁶²

[Fig. 7]

Fig. 7. DCMD tests of NIPS and PI-s5 membranes with different feed solutions: (a) 10 wt% NaCl; (b) 1000 ppm Rose Bengal dye in 10 wt% NaCl, where the flux values of NIPS and PI-s5 were normalized by their initial values of 12.3 kg/m² hr and 16.2 kg/m² hr, respectively; (c) 500 ppm humic acid in 10 wt% NaCl, with the flux values of NIPS and PI-s5 normalized by their initial values of 12.6 kg/m² hr and 15.0 kg/m² hr, respectively; (d) Different SDS concentrations in 10 wt% NaCl, where SDS was incrementally added to the feed tank every 2 hours until the SDS concentration reached 0.2 mM. All MD tests were conducted with feed solutions maintained at 60 °C, and initially, DI-water was used as the permeate and controlled at 15 °C. The solid and hollow symbols represent the vapor flux and rejection, respectively.

Comment 3. There is one contradictory in the Fig. 7(a). The NIPS membrane was wetted at about 3 hour, why the rejection decreased sharply at about 20 hour. It's unreasonable.

Reply: We thank the reviewer for pointing out this problem. We have performed the MD tests again for the NIPS membrane and made corrections to the corresponding paragraph on page 23, lines 514-518.

Comment 4. Also, the authors didn't give the explanation for the anti-wetting property for the PANIPS membrane. The authors should clarify where the self-cleaning and anti-wetting performance come from, the super-hydrophobicity or the piezoelectricity.

Reply: All self-cleaning and anti-wetting performances are due to the enhanced super-hydrophobicity after plasma treatment. To ensure clarity, we have further augmented our explanations on this matter as follows.

[Page 9, lines 205-208] ...The improved self-cleaning characteristic of PANIPS membranes can be primarily attributed to their increased surface roughness and porosity, which function similarly to the lotus effect in maintaining the cleanliness of the membrane surface.

Comment 5. As for the superhydrophobic property, the increasement of the roughness is vivid, presenting a remarkable increase in the water contact angle from about 95° to 166.4° and sliding angle. Please add XPS characterization to figure out if there is new functional group formed on the membrane surface.

Reply: Although the IR spectra did not reveal additional functional groups, we agreed with the reviewer that the XPS analyses could further verify if the elemental composition is unchanged after plasma treatment. We have conducted XPS analyses on the NIPS and PANIPS membranes as suggested. The survey spectra indicate that all PANIPS membranes are composed of C and F. The plasma treatment did not introduce any oxygen- or nitrogen-related functional groups to the membrane. The data and the explanation were provided in the revised manuscript and supplementary data as follows.

[Page 11, lines 248-254] ... *It is important to highlight that the PANIPS process exclusively alters the crystalline polymorphisms and morphology of the PANIPS membranes without introducing new chemical bonding. As revealed by the XPS survey spectra, the NIPS and PANIPS membranes only display characteristic peaks of C1s and F1s (Fig. S7a). A closer examination of the C1s narrow scan spectra further confirms that all membranes share common peaks of C-H (286.6 eV) and C-F (291.2 eV) with equivalent peak intensities (Fig. S7b).*

Comment 6. For the waveforms of the piezoelectric voltage output of NIPS and PANIPS membranes, where the output of the NIPS membrane come from? Since the NIPS membrane wasn't poled, it should not possess the piezoelectricity.

Reply: As revealed by ATR-FTIR, the PVDF membrane prepared by the NIPS method without electrical poling still contains 60-70% electroactive β phase. As a result, a weak voltage output can be generated and detected. A similar result was also reported by Choi et al. (<https://doi.org/10.1016/j.nanoen.2017.01.062>)

Comment 7. Page 20, Line 450-451, "the changes in piezoelectric output voltage are in accordance with the ratio of the electroactive phases and the total crystallinity." This sentence is totally wrong. The piezoelectric output is mainly related to the C-F dipole arrangement within the membrane matrix. The piezoelectricity should be evidenced by d33 test. d33 is a vital parameter for the piezoelectric material.

Reply: Thanks for the valuable comments. The d33 measurements were conducted and provided in the supplementary results (Fig. S11) as suggested. Overall, the d33 value increased with more scan cycles, which is in accordance with the voltage output experiments in Fig. 6.

Comment 8. Actually, the authors presented a new way for modifying the membrane fabrication and the work is heavy. However, the authors didn't organize the work very well. From my point of view, the presented self-cleaning property is caused by the rough surface, which is not related to the piezoelectricity. And for the self-cleaning and piezoelectricity, the authors didn't discuss in depth for the origin and the relevance.

Thus, this manuscript does not reach the level of Nature Communications.

Reply: Thanks for the valuable suggestions. The reviewer rightly notes that the efficacy of our self-cleaning tests is linked to the increased surface roughness and porosity aspects. To ensure clarity, we have further augmented our explanations on this matter. Our primary objective in this study is to introduce the PANIPS method as a novel approach for membrane fabrication. As pioneers in proposing this method, we

extensively characterized PANIPS membranes and provided insights into the underlying mechanisms governing the plasma treatment. We firmly believe our work not only offers comprehensive insights into PANIPS membranes but also serves as a guiding framework for utilizing plasma techniques to enhance membrane properties. Given our emphasis on understanding mechanisms, it's essential to highlight that our examination of piezoelectric properties and the DCMD tests primarily served to validate the superior performance of PANIPS membranes.

Reviewer #2 (Remarks to the Author):

The paper describes a new technique for membrane manufacturing that exploits an Ar microplasma to promote the phase separation process, enhancing the membrane surface roughness and favours the formation of piezoelectric PVDF phases. The manuscript describes thoroughly this new technique, showing the effects of several preparation parameters of the structure and performance of the membranes. This is clearly a high value research that propose a very innovative technique. Therefore, I think that the paper can surely be published on nature communications journal. However, out of pure curiosity, I would like the authors to answer some questions:

Reply: We thank the reviewer for taking time to review our manuscript and provide positive comments to improve the manuscript. Please find our point-to-point replies.

Comment 1. How long did the plasma take to treat the whole sample?

Reply: The total time required to treat 1 cycle with an area of $10 \times 12 \text{ cm}^2$ was about 35 s (under the conditions of the moving speed on the x- and y-axis of 20 mm/s and 50 mm/s, respectively). We have added this information to the experimental procedures as follows.

[Page 28, lines 611-613 ... The total time required for 1 scan cycle with an area of $10 \times 12 \text{ cm}^2$ was approximately 35 s, given the conditions of the moving speed on the x and y-axes at 20 mm/s and 50 mm/s, respectively. ...

Comment 2. After the treatment the sample was immediately immersed in water, or some time was required to do this operation?

Reply: We thank the reviewer for the important question. The plasma-treated membrane was immediately transferred to a water coagulant bath.

[Page 28, lines 613-615] ... After the plasma treatments, the membranes were immediately immersed in a tap-water coagulant bath to complete the phase inversion for 20 hr; ...

Comment 3. How difficult would it be to scale this setup to an industrial production.

Reply: The production of membranes through the NIPS method has reached a mature stage. We are confident that the scaling-up of PANIPS membranes in industrial production is generally not challenging due to the use of an 'atmospheric pressure' microplasma in the PANIPS method. The atmospheric pressure plasma is well-known for its seamless integration into the production line. Our study utilized a single

microplasma jet to scan membranes for cycles. To scale up, one may consider replacing the single plasma jet with a plasma array and controlling the scan speed to simulate different scan cycles. With proper optimization, it is possible to reach the same membrane performance presented in this work. A conceptualized continuous PANIPS membrane production line is illustrated in Fig. A1.

Fig. A1. An illustration of a conceptualized continuous PANIPS membrane production line.

Comment 4. On page 9 the authors state “... LEP because of the formation of the nodular structure and the weak mechanical properties of the spherulitic PVDF crystals.” I think that the problem could be the fragility of the connections between the spherulites more than the mechanical properties of the spherulites themselves. Did the authors evaluate the tensile strength at break of the membranes?

Reply: We thank and agree with the reviewer’s comments that the description is not accurate. We have rephrased the paragraph in the revised manuscript as follows. In addition, the tensile tests were conducted and provided in the supplementary file (Fig. S8).

[Page 13, lines 289-292] ...*However, despite the similar hydrophobicity of P1-s5 ~ P1-s9, a longer treatment cycle results in a lower LEP because of the formation of the nodular structure (Fig. 2 (d4) - (d6)). As shown in Fig. S8, the poor connection between the spherulitic PVDF crystals significantly reduces the mechanical properties.*

Comment 5. On page 15, the Ar flow rate is expressed in sccm. I suggest to use a clearer nomenclature such as cmstd/min.

Reply: We thank the review’s suggestion. In our device, the gas flow rate is regulated by a mass flow controller, employing the unit 'sccm' (standard cubic centimeters per

minute). Consequently, we adopt the same unit expression in our study. We thank the reviewer's suggestion to avoid misunderstanding. Upon careful review, we have verified the use of 'sccm' and 'cmstd/min,' and concluded that 'sccm' is a widely accepted unit, minimizing the potential for confusion.

Reviewer #3 (Remarks to the Author):

This manuscript reports discovery of a new approach for PVDF membrane fabrication based on plasma-assisted non-solvent induced phase separation. The resulting membrane is superhydrophobic, and is found to be quite effective when applied for direct contact membrane distillation (DCMD). The study is innovative and systematic, focused on illustrating the underlined membrane formation mechanisms. The manuscript is well prepared and easily to follow. Listed below are some minor comments that should be considered during the revision process.

Response: We thank the reviewer for taking time to review our manuscript and provide positive comments to improve the manuscript. Please find the following our point-to-point replies to the reviewer's comments.

Comment 1. Abstract: There appears to have no need to mention motion sensor, CO₂ capture etc.; those are not discussed in the main body of the paper.

Reply: We thank the reviewer's suggestions. We have removed them from the abstract as suggested.

Comment 2. P7 and figure 2: What is the duration for each cycle of atmospheric pressure plasma treatment? Plasm temperature?

Reply: The total time required to treat 1 cycle with an area of 10*12 cm² was about 35 s (under the conditions of the moving speed on the x- and y-axis of 20 mm/s and 50 mm/s, respectively). We measured the temperature of the membrane during the plasma treatment because it may affect the structure of the resulting membrane. As shown in Fig. S9, the temperature of the membrane only increases by 1 °C after 9 scan cycles. The information has been added to the revised manuscript as follows.

[Page 14, lines 313-314] *The effect of temperature increment is negligible, as the plasma treatment for 9 cycles only increases the membrane temperature by 1°C (Fig. S9). ...*

[Page 28, lines 611-613] *... The total time required for 1 scan cycle with an area of 10×12 cm² was approximately 35 s, given the conditions of the moving speed on the x and y-axes at 20 mm/s and 50 mm/s, respectively. ...*

Comment 3. The resolution of SEM images appears low.

Reply: We appreciate the reviewer's comment on this issue. The SEM images may have lost resolution due to compression. We have reviewed and substituted them with

higher-resolution versions.

Comment 4. Figure 6: Water clusters are considered important for the membrane development process, but there appears to be no direct evidence regarding the cluster population and sizes.

Reply: We appreciate the reviewer for the insightful question. The presence of water clusters should be beyond doubt, as the kinetics governing their formation in atmospheric pressure plasma systems have been extensively established and studied in the past decades. All relevant research indicates that water clusters constitute the ultimate products of atmospheric pressure plasma jets when water molecules are present in the environment. In our future work, we intend to collaborate with other research groups to investigate and establish correlations between the population or size of water clusters and the formation of PANIPS membranes.

Although lacking specific details on their population and size, we conducted two crucial analyses: Optical Emission Spectroscopy (OES) and measurements of the membrane's mass change after plasma treatment. Our OES results clearly demonstrate a positive correlation between humidity levels (RH 30-70%) and the formation of OH, H, O, and other reactive Ar species (Fig. 5c). These species are reported to promptly associate with water. Notably, we observed a decrease in the effectiveness of the plasma treatment as the intensity of these species reduced (Fig. 5b). Additionally, we observed a gradual increase in the membrane's weight with each treatment cycle, which could be considered direct evidence of water cluster deposition on the membrane.

Comment 5. P16: The statement “The charged water ion clusters are then accelerated by the electric field ...” is questionable. AC, not DC, is used for the plasma production, so the ions are not moving towards a specific direction under the electric field. Brownian motion could lead to attachment of water clusters to the dope liquid surface.

Reply: We acknowledge the reviewer’s point regarding the role of Brownian diffusion in water cluster attachment. We also believe that additional factors such as electric field acceleration, gravity, and convection might also play significant roles. Although an AC plasma is used, the charged species are still subject to the influence of the electric field which reverses at a fixed frequency (in our work would be 20 kHz), akin to the dynamics observed in the sputtering process. In our revised manuscript, we have updated the description as follows to encompass these plausible factors as potential reasons for water cluster attachment.

[Page 16, lines 343-344] ... *Additionally, gravity and Brownian diffusion may also facilitate the water deposition.*⁴⁹

[Page 16, lines 352-355] ... *Clearly, the formation of plasma and water clusters is the key to rendering the membranes with superhydrophobicity (Fig. 5b, P1-s9). While the Ar gas flow alone may not exhibit direct effectiveness, it likely aids convection, facilitating the transportation of water clusters to the membrane surfaces.*

[Page 17, lines 375-377] ...*These charged water ion clusters are further deposited on the membrane surface due to the electric field acceleration, gravity, convection, or Brownian diffusion, creating hierarchical nanostructures beneficial for the superhydrophobicity. ...*

Comment 6. P20/Figure 8: Piezoelectric phenomena for PVDF is not new. Is the V_{pp} observed there the highest when compared with PVDF prepared by other approaches?

Reply: In Table S2, we have compiled a summary detailing the performance of various piezoelectric PVDF membranes. Addressing the recommendation from reviewer 1, we conducted measurements to determine the d_{33} value of the PANIPS membranes. Remarkably, our findings indicate that both the output voltage and the d_{33} value of the PANIPS membranes are comparable to, or in some cases surpass, those of PVDF composite membranes utilizing graphene, carbon nanotubes, or other additives to produce membranes with a higher β phase.

REVIEWER COMMENTS

Reviewer #1 (Remarks to the Author):

The authors give a scientific and reasonable reply for my questions. There are small questions for the revision.

1. From Fig. 6, P1-S7 shows the biggest V_{pp} up to 10 V. But for the d33 measurement, the d33 increased with the scanning time. How much force is used during the test? Please explain why the P1-S7 shows the highest voltage output.
2. From the tensile test and the morphology, there should be a change in the pore size for P1-S5 and P1-S9. But in Fig. 7 and Fig. S12, both membranes have a similar flux about 20 LMH, please check.
3. Fig. 4, add the main peak, 840 cm^{-1} for β phase PVDF. Fig. S13, add the arrow to distinguish the flux and rejection rate.

Reviewer #2 (Remarks to the Author):

The revised manuscript is a great improvement of the previous version. The authors clarified all the points that were highlighted by the reviewer. I believe that the manuscript can now be published in Nature Communications as it is.

Reviewer #3 (Remarks to the Author):

I have reviewed the revised manuscript and believe the authors have adequately addressed reviewers' comments. These included their additional experiments on fouling evaluation and other characterizations. The paper is publishable.

All modifications made to the manuscript and supplementary information were highlighted in green. Below please find our point-by-point responses to each comment.

Reviewer's comments:

Reviewer #1 (Remarks to the Author):

The authors give a scientific and reasonable reply for my questions. There are small questions for the revision.

We thank the reviewer for the questions that helped to increase the clarity of the manuscript. The following lists our point-to-point replies.

Comment 1. From Fig. 6, P1-S7 shows the biggest V_{pp} up to 10 V. But for the d33 measurement, the d33 increased with the scanning time. How much force is used during the test? Please explain why the P1-S7 shows the highest voltage output.

Reply to comment 1. The force and the frequency used during the piezoelectric output voltage tests were 1 N and 1.82 Hz, respectively. On the other hand, they were 0.25 N and 110 Hz during the d33 measurements. The information is provided in section 1.3 in Supplementary Information.

We prepared new samples and re-tested the output voltage of P1-s7 and P1-s9 membranes again using newly prepared PANIPS membranes. As shown in Fig. A1 below, the output voltage of P1-s7 is confirmed to be higher than that of P1-s9.

Fig. A1. The piezoelectric output voltage of P1-s7 and P1-s9 membranes.

As demonstrated by Wang et al., the output voltage can be affected by many factors

such as compression area, membrane thickness, reciprocating frequency, and other piezoelectric coefficients [1]. On the other hand, the d_{33} value measured by the d_{33} tester is rather geometry-independent [2]. In our case, several possibilities could account for the decreased output voltage of P1-s9. For example, the thickness of the P1-s9 membrane is thicker than that of the P1-s7 membrane, where the longer distance between the electrodes could limit the electrostatic induction effect. Additionally, the structure and mechanical properties of the two membranes differ, implying that their resilience or the speed of elastic recovery could also change. Finally, the weak mechanical properties also lead to the membranes cracking easily, thereby affecting the actual compression area. Due to the membranes being sandwiched between two copper tapes for testing, assessing the membrane's condition is impossible. We believe that further in-depth research is necessary to determine the underlying reasons for this discrepancy, which exceeds the scope of the current study. We have included additional experimental details in the supplementary information and have also expanded on the potential causes for the varying trends in d_{33} and output voltage in the revised manuscript as follows.

[Section 1.3, Supplementary Information] *The piezoelectricity coefficient d_{33} was directly measured using a wide range d_{33} tester meter (APC International, Ltd.).⁸ The membrane samples were metalized by the silver paste from both sides with an area of 1×1 cm², followed by covering both sides with copper foils. The metalized sample was clamped between two metallic jaws with a static force; the position of the upper jaw was static during the measurement, while the bottom jaw was excited by a harmonic mechanical oscillation force with an amplitude of 0.25 N and a frequency of 110 Hz. Please check if my revisions are correct. At least three replicates of each membrane were measured to obtain the averaged d_{33} value.*

[Page 21, lines 477-483] *... Interestingly, despite P1-s9 having the highest d_{33} value, its output voltage is lower than that of P1-s7. This discrepancy might stem from the fact that while the d_{33} measurement is geometry-independent, the output voltage is influenced by various factors such as compression area, membrane thickness, reciprocating frequency, and other piezoelectric coefficients.^{58, 59} Therefore, a further in-depth study is required to assess the underlying reasons in the future.*

Comment 2. From the tensile test and the morphology, there should be a change in the pore size for P1-S5 and P1-S9. But in Fig. 7 and Fig. S12, both membranes have a similar flux about 20 LMH, please check.

Reply to comment 2. We verified the calculation and confirmed that the fluxes of P1-

s5 and P1-s9 are relatively similar, approximately 18-20 LMH. We also agree with the reviewer's comments that there should be potential variations in pore size across different plasma scan cycles. Therefore, we attempted to evaluate the pore size using a capillary flow porometer. As depicted in Table S3, the mean pore size consistently increases with the number of plasma scan cycles, correlating well with our observations from the SEM images. However, due to the weak mechanical properties of P1-s5 to P1-s9, we encountered difficulties in directly measuring their mean pore sizes. Nevertheless, one would anticipate the pore size of P1-s9 to be larger than that of P1-s5 based on the previous trend and the SEM images. That is to say, P1-s9 was expected to exhibit a higher flux rather than being at the same level as P1-s5.

During MD tests, the different flow rates of the feed and permeate solution result in the pressure gradient across the membrane. Given the weak tensile strength and the loosely connected nodular structure of the P1-s9 membrane, it's suspected that the membrane underwent compression and densification during MD measurements. Subsequently, the flux of P1-s9 decreased to a similar level of P1-s5. The discussion has been provided in the supplementary information as follows.

[Page S22, supplementary information] *According to the wettability tests, it is expected that PANIPS membranes with scan cycles >5 would all exhibit superhydrophobic properties with good fouling resistance. Moreover, as depicted in Table S3, the mean pore size consistently increases with the number of plasma scan cycles, which correlates well with the SEM images. However, due to the weak mechanical properties of P1-s5 to P1-s9, directly measuring their mean pore sizes becomes challenging. Nevertheless, an anticipated increase in pore size when prolonging the plasma scan cycle can be inferred from the trend observed in the SEM cross-sectional images. Hence, it is expected that the flux could be further enhanced by using the P1-s9 membrane.*

The P1-s9 membrane was then applied to treat both a 10 wt% NaCl solution and a solution containing 1000 ppm Rose Bengal dye. As demonstrated in Fig. S12, P1-s9 maintained a stable flux with rejections above 99%, regardless of the solution type. Its fouling resistance is also verified. However, with 10 wt% NaCl solution as the feed, the flux of the P1-s9 membrane is not higher but similar to that of the P1-s5 membrane, contradicting to our initial expectation. Considering the weak tensile strength and loosely connected nodular structure of the P1-s9 membrane, it is suspected that the membrane underwent compression and densification during the MD measurements. Consequently, the flux of P1-s9 decreased to a level akin to that of P1-s5.

Comment 3. Fig. 4, add the main peak, 840 cm⁻¹ for β phase PVDF. Fig. S13, add the arrow to distinguish the flux and rejection rate.

Reply to comment 3. Thank you for the suggestions. Fig. 4 and Fig. S13 have been adjusted as suggested. In Fig. 4, the peak at 840 cm⁻¹ was assigned to both β and γ phases [3]. Arrows were added to Fig. S13 to distinguish the flux and rejection.

Reference:

- [1] Wang, M., et al., Enhanced Nanogenerator of Embedding Lead-Free Double Perovskite Cs₂AgBiBr₆ in Polymer Matrix for Hybrid Energy Harvesting. *Journal of Materials Chemistry C*, 2023.
- [2] Stewart, M., W. Battrick, and M. Cain, Measuring piezoelectric d₃₃ coefficients using the direct method. 2001.
- [3] Cui ZL, Hassankiadeh NT, Zhuang YB, Drioli E, Lee YM. Crystalline polymorphism in poly(vinylidene fluoride) membranes. *Prog Polym Sci* 51, 94-126 (2015).

REVIEWERS' COMMENTS

Reviewer #1 (Remarks to the Author):

The manuscript is publishable now.